

**Historical Aurora Borealis Observations in Anatolia during medieval period:**
**Implications for the past solar activity**
Nafiz MADEN[a,1]
[a] Department of Geophysics, Gümüşhane University, TR-29100 Gümüşhane, Turkey
**Abstract:** In this paper, it is reviewed the relationships between the aurora
observations, past solar activity and climatic change in Anatolia during medieval period.
For this purpose, it is presented two historical aurora catalogs for Anatolia and Middle
East regions at various dates in order to understand the past solar activity and possible
physical mechanism using historical texts, chronicles and other auroral records. The
available catalogs in literature are covered records observed in the Europe, Japan,
China, Russia and Middle East. There is no study dealing only with the historical aurora
observations recorded in Anatolia. The data of the catalog strongly support that there is
a considerable relationship between the aurora activity and past strong solar activity. An
unusually high auroral activity during the years around 1100 in Anatolia and Middle East
is quite consistent with the past solar variability, geomagnetic field intensity and
planetary climatic changes drastically impacting on the economy and human events.
**Keywords:** Historical aurora records; Solar activity; Climatic changes; Anatolia.

[1] Corresponding author. Tel.:+90 456 233 74 25; fax: +90 456 233 74 27.
  E-mail: nmaden@gumushane.edu.tr (N. MADEN).



## 1.    Introduction

A number of researchers presented the low and middle-latitude aurora catalogs (Table 1) from Europe (Mairan, 1733; Frobesius 1739; Fritz, 1873; Schove, 1948; Link, 1962; Dall'Olmo, 1979; Stothers, 1979; Krivsky and Pejml, 1988; Vaquero et al., 2010; Scafetta and Willson, 2013), Arabic countries (Basurah, 2006), Japan (Matsushita, 1956; Nakazawa et al., 2004; Shiokawa et al., 2005), and China (Schove and Ho, 1959; Keimatsu,1976; Hayakawa et al., 2015). Aurorae are the most majestic luminous phenomenon observed in the sky. The historical aurora catalogs have been used to recognize the past solar activities (Siscoe, 1980; Silverman, 1992; Schröder, 1992; Schröder 1994; Basurah, 2006; Vazquez et al., 2006; Hayakawa et al., 2015), Earth's climate change (Pang and Yau, 2002; Schröder, 2004; Gallet et al., 2005; Bard and Frank, 2006; Scafetta, 2012) and perception of human civilizations (Schröder, 2004; Gallet et al., 2006; Silverman, 2006). The state of the geomagnetic field and the form of magnetosphere extremely control the location of auroral zone (Korte and Stulze, 2016). The visibility of the aurorae at low latitudes is very scarce and closely connected with the strong geomagnetic storms related to the high-speed solar wind or interplanetary transients (Eather, 1980; Basurah, 2006; Vazquez et al., 2006).

Mairan (1733) presented that the first scientific monography covers a list of 229 historical aurorae during the period of 502-1731. In I852, Wolf noticed that the aurorae match with periods of high sunspot number, according to the historical aurora catalog including more than 6300 records (Wolf, 1857). Fritz (1873), who listed 77 European Aurora records during 1707-1708, published the historical auroral catalog and separated auroral sightings into five categories based on the latitude and longitude (Schröder, 1994). Link (1962) published a useful aurora catalog seen in European countries based





on eight previous catalogs compiled by Frobesius (1739), Mairan (1754), Schoning
(1760), Boué (1856), Wolf (1857), Lovering (1868), Fritz (1873) and Seydl (1954).
Dall'Olmo (1979) listed 59 auroral observations displayed throughout medieval
period reported from European sources. Stothers (1979) compiled an extensive ancient
aurora catalog in Europe observed by the Mediterranean basin peoples and proved the
ancient auroral cycle simulates the modern cycle.
Vaquero et al. (2010) declared a set of auroral observation of Francisco Salva
Campillo who recorded in Barcelona during 1780-1825. This catalog represents a
sudden drop in the number of annual auroral observations at about 1793 owing to the
secular minimum in solar activity (Vaquero et al., 2010). Scafetta and Willson (2013)
studied the historical Hungarian auroral records covering 438 years. They found that the
maxima of the auroral observations conform to the maxima in the sunspot records and
there is a positive correlation amidst the auroral records, the solar and climate activities.
Korte and Stolze (2016) showed that the intensity and tilt of the geomagnetic field and
high solar activity are closely related to the Aurora occurrence.
The available catalogs described above present a number of records covering
Europe, Japan, China, Russia and Middle East. There is no study dealing only with the
historical aurora observations recorded in Anatolia. Anatolia have not been studied until
now with respect to meteorological and aurora observations. The goal of this study is to
compile a historical Anatolian aurora catalog (hAAc) during medieval period by scanning
the available sources and catalogs in literature. The catalog could be used to analyze
the past solar activity and planetary climatic changes impacting on the economy and
human events. This research may also contribute to the understanding of public
perception of the historical auroras.





**2.	Historical Anatolian Aurora Catalog (hAAC) through medieval period**

It is propounded a historical aurora catalog observed only in Anatolia during medieval period collected from Link (1962), Botley (1964), Baldwin (1969), Newton (1972), Stothers (1979), Eather (1980), Melissinos, (1980), Silverman (1998), Dall'Olmo (1979), Andreasyan (2000), Little (2007), Silverman (2006), Neuhäuser and Neuhäuser (2015) resources. In this catalogue, 23 different historical aurora records observed in Anatolia are presented during medieval period in Table 2. The location map of the historical Anatolian observations is given in Figure 1. A number of Anatolian aurora observations are summarized in Table 3. Another collected ancient aurora catalog consisting 45 auroral observations is shown in Table 4 for the Middle East region during the same period using Islamic historical texts, Arabic chronicles and other auroral records given in Table 1. These two catalogues are plotted in Figure 2 and evaluated altogether. The historical Anatolian and Middle East aurora records overlap through medieval period especially between 1097 and 1129 years (Figure 3). Also, Chinese and European aurora observations are in harmony with each other in this period (Siscoe, 1980).

According to the paper by Neuhäuser and Neuhäuser (2015), five criteria are implemented to perform the aurora catalogs as night-time (darkness, sunset, sunrise), non-southern directions (northern, NE, NW, E-W, W-E), color (red, reddish, fiery, bloody, green, black), dynamics (fire, fiery), and repetition. The strength of the aurora can be determined by considering its color, brightness, dynamics, duration, geomagnetic latitude. The observation is classified as potential (N=0), possible (N=1), very possible (N=2), probable (N=3), very probable (N=4), or certain (N=5) according to the criteria number (N) satisfied (Neuhäuser and Neuhäuser, 2015).



In Anatolia, the first auroral observation was done in Constantinople at 333
(Stothers, 1979). Stothers (1979) described these observations as a sky fire (N=1)
according to the works of Aurelius Victor (320-390), who was a historian and politician of
the Roman Empire. On the other hand, Eather (1980) described an Aurora observation
over Constantinople at about 360 BC during the siege on Byzantium by Philip of
Macedonia.
Little (2007) described an aurora observation record in Constantinople at 396: "A
fiery cloud was observed from the East while the city darkened. At first, it was small, but
later gradually grew and moved towards the city. At last, it terribly enlarged and poised
over the entire city. A terrifying flame appeared to hang down. All people stacked to the
church, and the place could not receive huge mass" (N=3).
According to the Link (1962), an aurora appeared in Asia Minor on 22 August
502, Thursday. This aurora was also observed both in Edessa (Botley, 1964) and
Palestine after an earthquake (Russell, 1985) based on to the Chronicle of Joshua the
Stylite. Joshua the Stylite described it: "On the 22rd of August this year, on the night
preceding Friday, a great fire appeared to us blazing in the northern quarter the all night.
It was believed that the whole earth was going to be devastated that night by a fire
storm. However, the mercy of our Lord preserved us without damage" (N=3). This
appearance of the aurora borealis was also reported in Chronicon Edessenum without
apocalyptic detail (Trombley and Watt, 2000).
According to the Historia Ecclesiastica of Ptolomaeus Lucensis there was an
aurora sighting at a night of 633 in Constantinople (Dall'Olmo, 1979): "A bloody sign
appearing just at that time was sighted. A bloodstained spear and a sharp light were
observed on the sky for nearly all night" (N=4). Theophanes (758/760-817), a Byzantine



monk, theologian, and chronicler, reported an observation in 667 winter: "There was a
sign which appeared in the sky in the same winter" (N=1). Theophanes reported another
observation in 675-676: "This year a sign was seen in the sky on a Sabbath day" (N=1;
Turtledove, 1982).
Theophanes recorded three aurora events for 734, 743 June and 744 in
Constantinople. The first aurora observation was reported in 734: "A fiery sign shining
like a burning brand appeared in the sky in Constantinople" (N=2). The second aurora
observation was recorded by Theophanes in June of 743: "In the northern sky of
Constantinople, a sign was observed in the month of June" (N=1; Turtledove, 1982). The
last aurora record was observed in Constantinople for 744: "In the northern sky, a sign
seemed this year, and dust fell in several places" (N=1; Turtledove, 1982; Neuhäuser
and Neuhäuser, 2015).
The low-latitude aurorae of 772-773 are interesting, as being very close to the
extreme solar event of 774-775 (Miyake et al., 2012; Usoskin et al., 2013; Mekhldi et al.,
2015). Harrak (1999) listed two aurorae records observed near Amida in the early 770s
based on the Chronicle of Zuqnin. In the Chronicle of Zuqnin, the first observation was
recorded in 772, Amida (Turkey): "Another sign was seen in the northern side, and its
view gave evidence about the menace of God against us. It appeared at reaping time,
while wrapping the whole northern side of the sky from west to east end. It was look like
a green sceptre, a red one, a yellow one, and a black one. It was ascending from the
ground and changing into 70 shapes, while one sceptre was emerging and another
disappearing" (N=3). The second observation was recorded in the Chronicle of Zuqnin in
773, Amida (Turkey): "In the month of June, on a Friday, another sign that was seen a
year ago in the northern region was appeared again this year. It was on Fridays that it





used to appear during these three consecutive years, stretching itself out from the
eastern side to the western side. The sign would change into many shapes in such a
way that as soon as a green ray vanished, a red one would appear, and as soon as the
yellow one vanished, a green would appear, and as soon as this one vanished, a black
one would appear" (N=3; Harrak, 1999; Neuhäuser and Neuhäuser, 2015). These two
observations listed by Harrak (1999) and Neuhäuser and Neuhäuser (2015) based on
the Chronicle of Zuqnin were also cited by Dall'Olmo (1979) according to the Chronique
de Denys de Tell-Mahré (Chabot, 1895) with different dating. Mekhaldi et al. (2015)
indicated that these two extreme events (772/773) were five times greater than any
other recorded solar storms with instruments. In Constantinople, another aurora
observation was recorded in 988: "A luminous star and fiery pillars seen in the northern
region of the sky for some nights. They frightened the people who saw them." (N=3;
Dall'Olmo, 1979).
Matthew of Edessa, who wrote a chronicle, described the events that occurred
between the years 952 and 1136, and reported four aurora observations around the
year 1100 (Andreasyan, 2000). Matthew of Edessa reported the first aurora observation
in the Armenian year 546 (25.02.1097–24.02.1098): "In this year, an odd and horrible
signs were observed in the the northern side of the sky. No one had ever seen such an
amazing omen so far. In the month of November, the sky kindled and reddened though
the air was clear and quiet. The bloody sky was covered with stacks as if clustered on
top of one another becoming colorful. The stacks were set to slip through in an easterly
direction, dispersed after having gathered, and enveloped the large amount of the sky.
Then, the dark redness such an amazing degree reached up to the middle of the sky
vault. The savants and sages interpreted this phenomenon that, it was a sign of



bloodshed. Actually, terrible events and disasters we included as a short story in our
book were soon to be fulfilled." (N=3).
Krey (1921) described an aurora observation during the siege of Antioch on the
account of eyewitnesses and participants in the first crusade: "A great earthquake
occurred on the third day before the Kalends of January (30 December 1097), and a
very fabulous sign was noticed in the sky. Northern part of the sky was so red that it
appeared as if sun rose to inform the day in the first sight of the night" (N=3). This
observation was also described by Baldwin (1969): "There was an earthquake on
December 30$^{th}$, and a frightening display of the aurora borealis next evening, and in this
way God chastised his army, so that we were intent upon the light which was rising in
the darkness, yet the minds of some were so blind and abandoned that they were
recalled neither from luxury nor robbery. At this time the Bishop prescribed a fast of
three days and urged prayers and alms, together with a procession, upon the people;
moreover, he commanded the priests to devote themselves to masses and prayers, the
clerics to psalms". On the other hand, another aurora was observed on 3 June 1098 at
Antioch based on the Link (1962) catalog as a fiery red sky (N=2; Silverman, 2006).
The Matthew of Edessa recorded second aurora observation in the Armenian
year 547 (25.02.1098–24.02.1099). "In the same year, a new sign appeared in the
northern part of the sky. At the fourth hour of the night, the sky appeared more inflamed
than before, and a dark red color. This phenomenon lasted from the evening until the
fourth hour of the night. Such a terrible omen had never been seen so far. This omen
raised upwards gradually and covered the northern portion of the sky with the lines
reaching the hills. All stars took a fiery color. This phenomenon was an omen of rage
and catastrophe" (N=4; Andreasyan, 2000). Botley (1964) reported an auroral



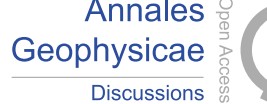

observation in Antioch as a blaze of light girdled Pole (N=1). Link (1962) dated this
observation on September 27, 1098.

In the Armenian year 548 (25.02.1099–24.02.1100) Matthew reported another

aurora observation: "A fiery sign of dark red color appeared in the sky in this year. This
omen heading from the northern to the eastern part of the sky appeared until the
seventh hour of the night and then became black. It was said that this phenomenon was
a sign of bloodshed of Christians. These predictions were truly realized. No favorable
omen did not appear since the day when the Franks began their expedition. All omens,
however, marked to realize the destruction, death, slaughter, famine and other diverse
disasters" (N=3; Andreasyan, 2000).

Matthew recorded the last aurora observation in the Armenian year 549

(25.02.1100–24.02.1101): "The northern part of the sky flushed red for the fourth time in
this year. The fiery red omen appeared more horrific than the previous one and
subsequently changed into black. This fourth appearance coincided with a lunar eclipse.
This phenomenon was a sign of the celestial wrath of God over the Christians as
previously said by the prophet Jeremiah with these words: "His wrath will blaze up from
the northern part of the sky. Indeed, several misfortunes occurred as we never could
have expected" (N=3; Andreasyan, 2000).

Dall'Olmo (1979) reported an aurora observation based on the Chronicle of

Michael the Syrian translated into French by Chabot (1968): "In the year 1108, a light
like the sunlight was seen in the middle of the night, and remained about three hours in
Djihan region near Adana" (N=2). Dall'Olmo (1979) was also cited 12 auroral records
observed probably in the Middle East from 745 to 1141 (Table 4) according to the
Chronicle of Michael the Syrian (Chabot, 1968).



On December 16, 1117, an aurora was recorded in Asia Minor (Link, 1962). In the
same date, two observations were also reported in the Middle East (Newton, 1972) and
in Palestine (Botley, 1964). These two observations could be same event. Link (1962)
described other observations in Asia Minor in the year 1119. This event might be the
same record observed in Armenia (Botley, 1964) given in Table 4.
Priest Grigor, who continued the Matthew's Chronicle and recorded events for the
years 1136/37-1162/63, added one aurora observation in about the year 1143. In the
Armenian year 592 (14.02.1143-13.02.1144) Priest Grigor described the aurora
observation: "On Holy Thursday (1 April 1143), an omen forming of a luminous column
appeared in the northern portion of the sky. This omen was visible for eight days. Three
sovereigns died after the appearance of this phenomenon" (N=3; Andreasyan, 2000).

**3.      Results and Discussions**
The main purpose of this study is to present an aurora catalog for the Anatolia
during the medieval period. 23 different historical aurora records are presented during
the medieval period in Anatolia (Table 2). Another aurora catalog containing 45 records
collected from different sources is also given (Le Strange, 1890; Link, 1962; Botley,
1964; Newton, 1972; Dall'Olmo, 1979; Silverman, 1998; Basurah, 2006) for the Middle
East region (Table 4). The aurora observations were described as "sign", "a fiery shining
sign", "a very fabulous sign", "red sky", "a fiery red sky", "sky fire", "a great fire", "a fiery
cloud", "a frightful and strange omen", "a fire-like omen", "a bloody spear light", "blaze of
light", "a sunlight light". The form of aurorae was defined as "luminous column". The
aurorae were generally seen in the northern and eastern part of the sky. The color of the
aurora observations were red, green, yellow and black depending on the height and





relative concentrations of the nitrogen and oxygen compounds in the atmosphere (Eather, 1980). The number N sort out only the probability that an event could be an aurora or not. The possibility of the aurora could be decided by regarding its duration geomagnetic latitude, color, brightness and dynamics. Aurorae observations with N≥3 tend to be true.

The aurora records strongly correlated to high solar activity (Siscoe, 1980) provide some information about the Sun-Earth interaction as previously proved by Scafetta (2012). They are the longest direct observational records available for studying solar and space weather dynamics. Stronger solar dynamics were realized in aurorae with color green-yellow-red as seen in 772 and 773 in Amida. Miyake et al. (2012) and Usoskin et al. (2013) confirmed the 770s high solar events presenting $^{14}$C measurements from the annual rings of the cedar trees in Japan and inappropriate carbon cycle model in German oak, respectively. The auroral records have also proven itself to be a valuable data source for the investigation of the secular variation of solar activity. Paleomagnetic researchs demonstrate that the recent dipole strength was nearly 50% weaker than it was 2500 years ago (Raspopov et al., 2003). Siscoe and Siebert (2002) indicated that the dipole strength was 1.5 times as large as that of the present value. The long-term variation of the geomagnetic latitude and dipole moment might be the reason of observing aurorae in Anatolia. The average dipole moment for 750 and 1250 are 8.85 $10^{22}$ Am$^2$ and 8.90 $10^{22}$ Am$^2$ slightly higher than the present value of 7.78 $10^{22}$ Am$^2$ (Korte and Constable, 2005; Gallet et al., 2005). According to the Kawai et al. (1965) the axis of geomagnetic dipole could have inclined towards Asia at around the 11$^{th}$-12$^{th}$ centuries. In addition, the possibility of auroral occurrence at low latitudes could demonstrate changes in the location of the North magnetic pole





(Silverman, 1998). This study could also be significant constraints for exploration of solar
activity on Earth's atmosphere and climate during the historical periods previously
proved by Bard and Frank (2006). According to the Bard and Frank (2006) solar
fluctuations caused climatic changes called Medieval Warm Period (900–1400). The
Maunder Minimum (1645-1715) which delineates the coldest part of the Little Ice Age
(Eddy, 1976) is depicted by a solar activity reduction, as well as a sunspots scarcity. A
new low sunspot number and lower aurora activity, which occurring in the period
between 2014 and 2025 (Li et al., 2018), might have led to a temporary change in
natural environment influencing the general public's attitudes and socio-economic
factors. Also, resource scarcity and disparities could also lead to social tensions in the
communities for the next ten years.

The position of the magnetic poles is the most important factor defining whether

the aurora was observed at a geographic region. Palaeomagnetic data provides similar
longitude values (85° N, 115° E) for the north geomagnetic pole (Merrill and McElhinny,
1983). The positions of the north magnetic pole have changed from 10° to 358° in
longitude and between 79° and 88° in latitude over the past 2500 years (Ohno and
Hamano, 1992). During the interval of 1127–1129, the north geomagnetic pole was
located at a geographic latitude of 80° N, and geographic longitudes including East Asia
(Merrill and McElhinny, 1983; Constable et al., 2000). According to the Fukushima
(1994), the north magnetic pole was located at 81°N in the eastern hemisphere near
East Asia (100°E to 130°E) in the medieval period. The north geomagnetic pole of dipole
axis computed from the average spherical harmonic models were 84.8° N and 103.8° E
in 1100 (Constable et al., 2000).





The geomagnetic latitude of Amida (Turkey) in the late 8[th] century to be about
50.1° (Neuhäuser and Neuhäuser, 2015) based on the Holocene geomagnetic field
(Nilsson et al., 2014) and 45° (Hayakawa et al., 2017) based on the location of the North
Geomagnetic Pole over the past 2000 years (Merrill and McElhinny, 1983). According to
the Silverman (2006), the geomagnetic latitude of Edessa and Antioch was 41° and 40°,
respectively. Strong geomagnetic storms, indicating strong solar activity around 770 and
1100 should have been exist in Amida (45°), Edessa (41°) and Antioch (40°).
The Medieval Climate Anomaly characterizing by warmer and drier climate
conditions generally related to relatively prolonged solar activity during the 12[th] and 13[th]
centuries (Jirikowic and Damon, 1994). Damon and Jirikowic (1992) estimated that the
rise of global temperature maxima stays below 0.8°C and anomalously high
temperatures pursue during the 12[th] and 13[th] centuries. Sharma (2002) revisited the
issue and proposed that very large solar variations have modulated climate over the
past 200 millennia. Gallet et al. (2006) demonstrated that fluctuations in the
geomagnetic field might trigger significant climate change impacting on some major
societal events in the Middle East at longer time. An inverse relationship amidst the
aurora records, severe winter and famine is estimated during the years of 1100 in
Anatolia. The high aurora activity could be reason of temperature rise during the
medieval period in Anatolia.
Haldon et al. (2014) subdivided Medieval into four climatic phases as dry (270-
540), very wet (540-750), moderately dry (750-950) and moderately wet (950-1400)
depending on archaeological, environmental, climate, high resolution pollen and stable
isotope data from sites in central and northwestern Turkey. However, this subdivision
should be revised as dry (0-560), very wet (560-725), moderately dry (725-990) and





moderately wet (990-1400) as given in Table 5 by using Anatolian and Middle Eastern
aurora observations besides meteorological data. Affective cold winter, wet climate
conditions, drought and famine could be occurred for Asia Minor and Middle East region
during 990-1400. It seems that the relatively high auroral activity during the years around
1100 both in Anatolia and Middle East indicates that solar activity must have been
intense rather than moderate causing the climate warmer (Fig. 2). In this period, Islamic
world was converted into an enlightened center for science, education, medicine, and
philosophy as previously stated by Hamilton (1982). An important increase in agricultural
production and population seems to have occurred in Anatolia after the year of 1100
where the aurora observations are intense (Fig. 2). Vaquero and Trigo (2012) stated the
period from 1095 to 1204 as an average solar cycle length. Bekli et al. (2017)
demonstrated that the naked-eye sun spot observations from 974 to 1278 and aurora
records from 965 to 1273 show multiple unusual peaks related to the high solar activitiy
at latitudes below 45° by using Chinese and Korean historical sources.
In the medieval period, the people were thought that the aurora was a sign of
anger of God, menace, threat, apocalyptic, doomsday, misfortunes, war, slaughter and
blodshed. Little (2007) described an aurora observation record in Constantinople at 396:
"All people stacked to the church, and the place could not receive huge mass. But after
that great tribulation, when God had accredited His word, the cloud began to diminish
and at last disappeared. The people, freed from fear for a while, again heard that they
must migrate, because the whole city would be destroyed on the next Sabbath. The
whole people left the city with the Emperor; no one remained in his house. The city was





saved. What shall we say? adds Augustine. Was this the anger of God or rather His
mercy"?
In the Chronicle of Zuqnin, an aurora observation recorded in 772, Amida
(Turkey) was described: "Another sign was seen in the northern side, and its view gave
evidence about the menace of God against us. For the intelligent person the sign
indicated menace. Many people said many things about it; some said it announced
bloodshed, and others said other things. But who knows the deeds of the Lord"?
Matthew of Edessa described the aurora phenomenon as a sign of rage,
catastrophe, and celestial wrath of God over the Christians and bloodshed of Christians.
Matthew of Edessa reported: "These predictions were truly realized. No favorable omen
did not appear since the day when the Franks began their expedition. All omens noticed
to realize the destruction, death, slaughter, famine and other diverse disasters"
(Andreasyan, 2000).

**4.    Conclusions**
This study establishing the strong solar activity during medieval period reports the
aurora observations recorded both in Anatolia and Middle East region integrating
meteorological data. The following conclusions can be summarized as follows:
1. Historical Anatolian aurora catalog (hAAc) containing 23 different aurora records provide
important information on variations in geomagnetic and auroral activity during medieval
period.
2. In Anatolia and Middle East, there was a relatively high auroral activity during the years
around 1100 is quite consistent with the naked-eye sunspot observations.

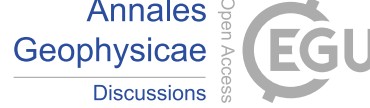

3.  *The historical Anatolian Aurora catalog exceptionally promote that there is a remarkable*

*correlation between the past solar activity and aurora activity.*

4.  The intensity of dipole moment and position of the geomagnetic pole might be the most

important factors observing aurorae in Anatolia and Middle East regions during medieval

period.

5.  In the Medieval period, Four climatic phases portrayed by Haldon et al. (2014) is revised

as dry (0-560), very wet (560-725), moderately dry (725-990) and moderately wet (990-

1400) depending on aurora observations besides meteorological data.

6.  Further investigations are required to establish a relationship between the solar variability

and climatic changes, such as the Medieval Climate Anomaly or Little Ice Age.

7.  People in medieval Anatolia were believed that the aurora was a sign of celestial wrath of

God, menace, threat, apocalyptic, doomsday, misfortunes, war, slaughter, rage,

catastrophe and bloodshed.


## 5.   Acknowledgements

We thank Elif KARSLI (KTU), Alam KHAN (GU) and anonymous referees for their

thorough critical and constructive comments. The authors are grateful to Editor for his
advice to improve the quality of this manuscript.

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



**TABLES CAPTIONS:**

**Table 1.** Historical Aurora catalogs compiled by different authors.

**Table 2.** Historical Anatolian Aurora catalogs during medieval period compiled in this study.

**Table 3.** The number of historical aurora records observed in Anatolia.

**Table 4.** Ancient aurora observations recorded in Middle East region during medieval period.

**Table 5.** Summary of Ancient climate change based on the aurora observations and meterological data in Anatolia during medieval period.

**FIGURE CAPTIONS:**

**Figure 1.** The location map of the historical Anatolian records during medieval period.

**Figure 2.** Comparison of historical aurora observations with climate change and meteorological data in Anatolia and neigbouring regions. The upper panel shows the meteorological data climatic subdivisions, the middle panel shows the aurora observations in Anatolia and Middle East regions and the lower panel shows the land use and population in Anatolia. Meteorogical and land use data are taken from Haldon et al. (2014).

**Figure 3.** The number of aurorae records per century observed in the Anatolia and in Middle East.





**TABLES**

**Table 1.**

| Sources | Number of Observations | Region | Period |
|---|---|---|---|
| Link, 1962 | 385 | Europe | 626 B.C. to 1600 A.D. |
| Link, 1964 | 209 | Europe | 1600-1700 A.D. |
| Stothers, 1979 | 67 | Greece and Italy | 480 B.C. to 333 A.D. |
| Newton, 1972 | 65 | Europe | 450-1263 A.D. |
| Dall'Olmo, 1979 | 61 | Europe | 450-1461 A.D. |
| Keimatsu, 1976 | 260 | China, Korea, and Japan | 687 B.C. to 1600 A.D. |
| Matsushita, 1956 | 18 | Japan | 620-1909 A.D. |
| Basurah, 2006 | 18 | Arabia, North Africa, Spain | 800-1600 A.D. |
| This Study | 23 | Anatolia | 1-1453 A.D. |
| This Study | 45 | Middle East | 1-1453 A.D. |











**Table 2.**

| # | Date | Location | Description | N | References |
|---|------|----------|-------------|---|------------|
| 1 | 333 | Constantinople | Sky fire. | 1 | Stothers, 1979 |
| 2 | 396 | Constantinople | A fiery cloud was seen from the East. | 3 | Little, 2007 |
| 3 | 22 Ağustos 502, Thursday | Edessa | A great fire appeared to us blazing in the northern quarter the whole night. | 3 | Link, 1962 |
|   |   |   |   |   | Botley, 1964 |
| 4 | 633 | Constantinople | A bloody spear and a light of the sky were sighted for nearly the all night. | 4 | Dall'Olmo, 1979 |
| 5 | 668 | Constantinople | There was a sign appeared in the sky in the same winter. | 1 | Turtledove, 1982 |
| 6 | 675 | Constantinople | In this year, a sign was seen in the sky on a Sabbath day. | 1 | Turtledove, 1982 |
| 7 | 734 | Constantinople | There was a sign in the sky which shone like a burning brand. | 2 | Turtledove, 1982 |
| 8 | June 743 | Constantinople | In June, a sign appeared on the northern sky. | 1 | Turtledove, 1982 |
| 9 | 744 | Constantinople | This year, a sign appeared on the northern sky. | 1 | Turtledove, 1982 |
| 10 | 772 | Amida | Another sign appeared in the northern side. | 3 | Harrak, 1999 |
| 11 | June 773, Friday | Amida | The sign that was seen a year ago in the northern region was seen again in this year, in the month of June, on a Friday. | 3 | Neuhäuser and Neuhäuser, 2015 |
|   |   |   |   |   | Harrak, 1999 |
| 12 | 988 | Constantinople | Frightened fiery pillars seen in the northern region for some nights. | 3 | Dall'Olmo, 1979 |
| 13 | 21 November 1097, Monday | Edessa | A frightful and strange omen appeared in the northern portion of the sky. | 3 | Link, 1962 |
|   |   |   |   |   | Silverman, 2006 |
|   |   |   |   |   | Andreasyan, 2000 |
|   |   |   |   |   | Botley, 1964 |
| 14 | 30 December 1097, Friday | Antioch | A very fabulous sign was watched in the sky. | 3 | Silverman, 1998 |
|   |   |   |   |   | Baldwin, 1969 |
|   |   |   |   |   | Botley 1964 |
|   |   |   |   |   | Kery, 1921 |
| 15 | 3 June 1098, Saturday | Antioch | A fiery red sky was seen. | 2 | Link, 1962 |
|   |   |   |   |   | Silverman, 2006 |
|   |   |   |   |   | Botley 1964 |





| | | | | | |
|---|---|---|---|---|---|
| 16 | 27 September 1098, Monday (10:00) | Edessa | A second omen appeared in the northern portion of the sky at the fourth hour of the night the sky flared up more than it had before and turned a deep red color. | 4 | Andreasyan, 2000 |
| | | | | | Link, 1962 |
| 17 | 27 September 1098, Monday | Antioch | Blaze of light girdled Pole. | 1 | Link, 1962 |
| | | | | | Botley, 1964 |
| 18 | 1099 | Edessa | A fire-like omen of a very deep red color appeared in the sky. | 3 | Andreasyan, 2000 |
| | | | | | Link, 1962 |
| | | | | | Silverman, 2006 |
| 19 | 18 November 1100, Sunday | Edessa | The northern portion of the sky reddened, appearing more frightful and wondrous than the previous phenomenon. | 3 | Andreasyan, 2000 |
| | | | | | Silverman, 2006 |
| | | | | | Link, 1962 |
| 20 | 1108 | Adana | A light like the sunlight was seen in the middle of the night, and remained about three hours in Djihan. | 2 | Chabot, 1968 |
| | | | | | Dall'Olmo, 1979 |
| 21 | 16 December 1117, Monday | Asia Minor | | | Link, 1962 |
| | | | | | Newton, 1972 |
| 22 | 1119 | Asia Minor | | | Link, 1962 |
| 23 | 1 April 1143, Thursday | Edessa | A sign appeared in the sky from the north in the form of a luminous column | 3 | Andreasyan, 2000 |





**Table 3.**

| # | City | Latitude [Degree, N] | Longitude [Degree, E] | Numbers of observation |
|---|------|----------------------|------------------------|-------------------------|
| 1 | Constantinople | 41.03 | 28.99 | 9 |
| 2 | Edessa | 37.17 | 38.79 | 6 |
| 3 | Amida | 37.93 | 40.21 | 2 |
| 4 | Antioch | 36.2 | 36.16 | 3 |
| 5 | Adana | 36.99 | 35.34 | 1 |
| 6 | Asia Minor | 39.93 | 32.85 | 2 |
| | | | Total | 23 |






**Table 4.**

| # | Date | Place | Decriptions | References |
|---|------|-------|-------------|------------|
| 1 | 65 | Jerusalem | | Botley, 1964 |
| 2 | 66 | Jerusalem | | Botley, 1964 |
| 3 | 400 | Byzantium | | Link, 1962 |
| 4 | 402 | Byzantium | | Link, 1962 |
| 5 | 473 | Byzantium | | Link, 1962 |
| 6 | 474 | Byzantium | | Link, 1962 |
| 7 | 502 Agust 22 | Palestine | A great fire appeared to us blazing in the northern quarter the whole night | Botley, 1964 |
| 8 | 743 June | Syria | A mighty sign appeared in the heavens like columns of fire blazing in June | Chabot, 1968 |
| 9 | 743 September | Middle East | Another sign appeared in September like a flame of fire and spread from the East to the West | Cook, 2001 |
| 10 | 745 January | Middle East | In the middle of the sky, a large column of fire appeared during the night | Chabot, 1968 |
| 11 | 793 May 11-17 | Iraq | There occurred a violent wind and overshadowing of the heavens and a redness in the sky, on the night of Sunday | Basworth, 1989 |
| 12 | 817 October 29 | Iraq | A reddish glow appeared in the sky and stayed until late at night like a two red columns | Basurah, 2006 |
| 13 | 840 September 24 | Middle East | A fiery cloud appeared in the northern part of the sky, moving from east to West. | Dall'Olmo, 1979 |
| 14 | 931 November 9 | Baghdad | An intense red glow appeared in the city of Al-Salam (Baghdad) | Basurah, 2006 |
| 15 | 939 October 17 | Syria | An intense red glow appeared in the atmosphere coming from North and West | Basurah, 2006 |
| 16 | 1050 Agust 5 | Middle East | Through which light shone out broad and glittering, and then became extinguished | Le Strange, 1890 |
| 17 | 1097 | Palestine | | Botley, 1964 |
| 18 | 1100 | Palestine | | Botley, 1964 |
| 19 | 1102 | Palestine | | Botley, 1964 |
| 20 | 1106 | Syria | | Botley, 1964 |
| 21 | 1110 | Syria | | Botley, 1964 |
| 22 | 1117 December 16 | Palestine | | Newton, 1972 / Botley, 1964 |
| 23 | 1119 | Armenia | | Botley, 1964 |
| 24 | 1121 May, Monday | Syria | There appeared a full arc, which had not been observed for many enerations | Botley, 1964 |
| 25 | 1129 January | Middle East | A fire appeared in the northern region. A sort of pillar was stretched toward the south. | Dall'Olmo, 1979 |
| 26 | 1129 March | Middle East | A fire appeared in the northern region. A sort of pillar was stretched toward the south. | Dall'Olmo, 1979 |






| | | | | |
|---|---|---|---|---|
| 27 | 1129 April | Middle East | A fire appeared in the northern region. A sort of pillar was stretched toward the south. | Dall'Olmo, 1979 |
| 28 | 1130 November | Middle East | A burning fire was seen in the northern region | Dall'Olmo, 1979 |
| 29 | 1135 July 21 | Middle East | A light like a torch moved from east to West. The light of the moon and of the stars was obscured. A frightful noise followed | Dall'Olmo, 1979 |
| 30 | 1138 October | Syria | A red sign was seen in the northern part of the sky | Botley, 1964 |
| 31 | 1140 June 22 | Syria | Red lances were seen in the northern region. | Botley, 1964 |
| 32 | 1141 August | Middle East | Rays of fire were observed in the northern region. | Dall'Olmo, 1979 |
| 33 | 1141 September | Syria | A brightness as bright as the sun broke out in the northeast. It shone as if the sky were on fire. | Botley, 1964 |
| 34 | 1149 | Syria | | Botley, 1964 |
| 35 | 1150 | Palestine | | Botley, 1964 |
| 36 37 | 1176 September 6 - October 5 | Syria | An intense red light appeared in the sky from the East | Basurah, 2006 |
| | 1179 May 7 | Syria | The sky became cloudy and pillars of fire appeared at the horizon | Basurah, 2006 |
| 38 | 1182 | Byzantium | | Link, 1962 |
| 39 | 1187 July | Tiberias, Israel | | Botley, 1964 |
| 40 | 1223 October 26 | Syria | We saw from Bani Helal Mountain (toward the North direction) a hugelight over Gassune; we thought that Damascus was on fire. | Basurah, 2006 |
| 41 | 1264 July 20–30 | Syria | A bright glowing columns appeared toward North-West | Basurah, 2006 |
| 42 | 1370 November 27 | Jerusalem | A great reddish glow appeared in the sky of Jerusalem | Basurah, 2006 |
| 43 | 1370 November 27 | Damascus | A great reddish glow appeared in the sky of Damascus | Basurah, 2006 |
| 44 | 1370 November 27 | Homs | A great reddish glow appeared in the sky of Homs | Basurah, 2006 |
| 45 | 1370 November 27 | Aleppo | A great reddish glow appeared in the sky of Aleppo | Basurah, 2006 |










**Table 5.**

| Medieval Period | | Climate |
|---|---|---|
| Haldon et al. (2014) | This Study | |
| 270-540 | 0-560 | Dry |
| 540-750 | 560-725 | Very wet |
| 750-950 | 725-990 | Moderately dry |
| 950-1400 | 990-1400 | Moderately wet |














**Figures**



**Figure 1.**





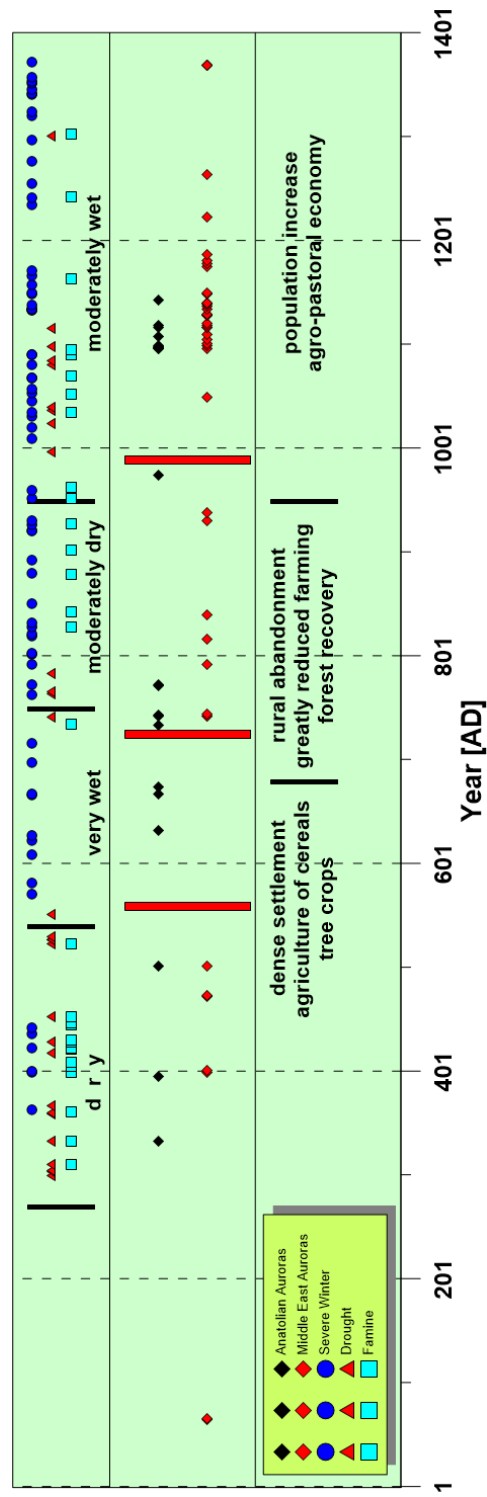

**Figure 2.**






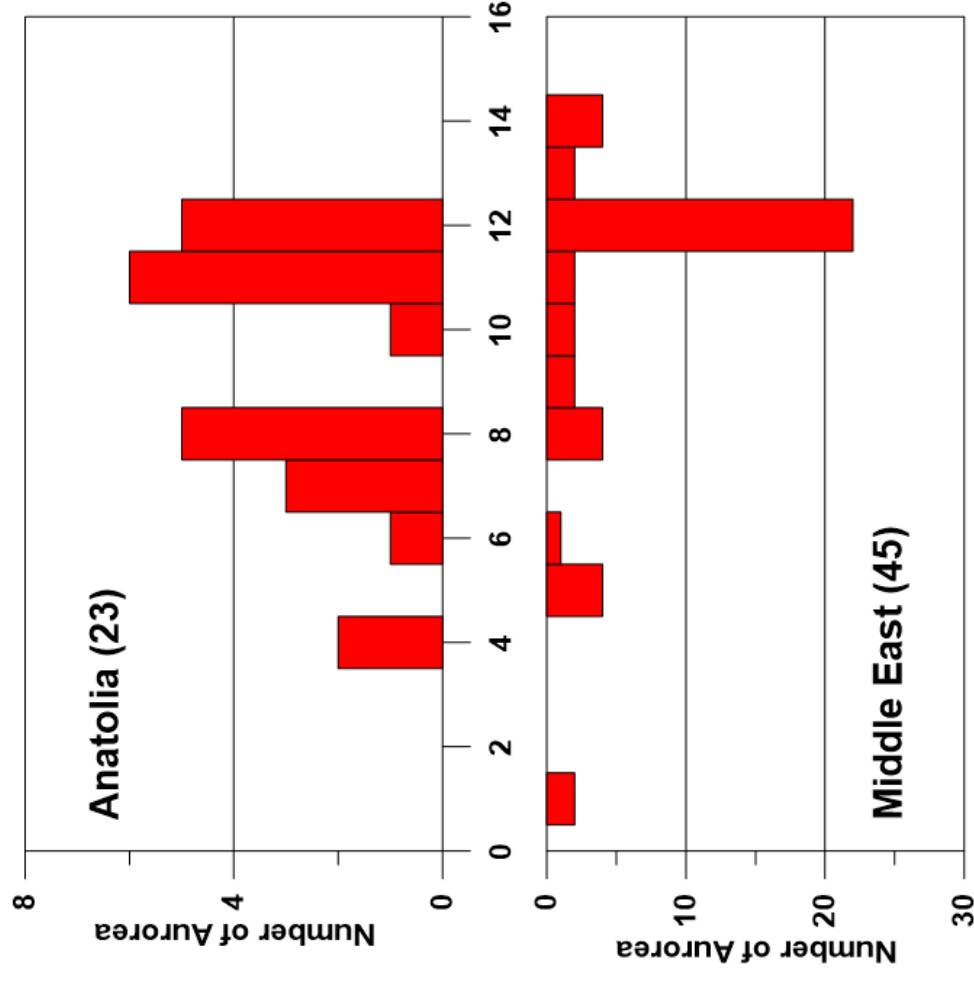

**Figure 3.**
