# Peer review of "Historical Aurora Borealis Observations in medieval Anatolia (AD 1-1453): Implications for the past solar activity"

_Annales Geophysicae, 2019_

## Referee Comment (RC1) · Anonymous Referee #1 · 10 Sep 2019

Review Manuscript Number: angeo-2019-97

Title: "Historical Aurora Borealis Observations in Anatolia during medieval period: Implications for the past solar activity"

Author: Nafiz Maden

**General Comments:**

In this paper, the author reports an overview of historical Aurora observations reports in Anatolia and Middle East regions in the medieval period based in historical texts, chronicles and aurora catalogs records. The paper tried to make a relationship between the auroral activity and the past solar activity, the past climatic changes, economy and society living in the remote time.

My view on the paper is that though the discussions are interesting, but the paper do not bring clear new results and it was missing a interconnection between the historical facts, the beliefs of the ancient people and the new science that explain the Aurora phenomenon as a direct relation between the Sun events and the geomagnetic field, and also the current status of aurora observation in the north and south hemispheres. There are some scientific explanation that should be present in the paper and the cited time periods in the medieval era need to be more clear, explaining/discussing a little more some sentences and not just cite the previous papers or historical texts/chronicles. Due to these problems found in the paper I recommend to not accept the paper as it is presented

**Major Comments:**

1) The title and other citation along the text do not include the approximate time period: e.g., in the title "Historical Aurora... during the medieval period...". I think that the approximate years or century could be cited. The correct location in time (type of calendar, year and/or century) should appear clearly in a historical text.

2) In the Introduction section it was not defined/explained the Aurora phenomenon, neither the physical mechanisms responsible for the occurrence of this event (solar particle precipitation/solar wind, geomagnetic storms, and loss cone of particle perception / Earth magnetic field configuration, and the interaction of such charged/energetic particles with the neutral/ionized compounds of the upper atmosphere); for this kind of paper would be very interesting to show a couple examples of Auroras registered in the present time; it was missing the explanation that Auroras exist in both hemispheres (North: Borealis Aurora; South: Austral Aurora), and the physical process/mechanism involved in the Aurora light emission is exactly the same, the magnetosphere and the Earth magnetic field configurations (approximately a magnetic dipole) and intensity have a very important role in the occurrence of Auroras (this was mentioned in some way, but not discussed and none map/cartoon was showed - for the present period). Auroras also can be observed in other planets besides Earth. In fact, I missed a deeper technical revision (including photos, a global map showing the Auroral regions in both north and south poles) in the introduction section.

3) In the Results and Discussion, I could not see discussion based on the summarized parts of historical records, and neither explanations of the relationship between Auroras and Climatic Changes/solar variability/society economy. These relations should be better discussed and explained in light of the current time when the climactic changes are discussed in a global scenario.

4) In the Conclusions, it was stated that in the paper was established a relationship between the strong solar activity and auroral activity by integrating meteorological data (pg 15, lines 347-349). I could not see any meteorological data and evidence for such relationship along the paper or this was not stressed

or adequate explained. On page 16, lines 363 and 364, the author suggested future investigations in order to establish a relationship between the solar variability and climatic changes. The current paper have the aim to obtain some relationship between these two phenomena, but it was not clear. Why the author do not use one of the suggestions (for example: "Medieval Climate Anomaly") and improve the current paper? This would be much more interesting than just make a revision on previous paper and texts from historical manuscripts, without a deep discussion.

**Minor issues:**

Line 10/11 (pg 1): "...in order to understand the past solar activity and possible physical mechanism using historical texts, chronicles and other auroral records?". At the end of the reading it was not clear/understandable the physical mechanism beyond the auroras. Can the author improve the paper in order to satisfy this purpose?

Line 14/15 (pg 1): "The data of the catalog strongly support that there is a considerable relationship between the aurora activity and past strong solar activity". Again, the paper did not clarify the above relationship.

Line 16-18 (pg 1): "An unusually high auroral activity during the years around 1100... is quite consistent with the past solar variability, geomagnetic field intensity and planetary climatic changes". The text did not present clearly the relationship between the unusually high auroral activity around 1100 and the planetary climatic changes.

Line 58 (pg 3): "... maxima of auroral observations conform to the maxima in sunspot records..." . The author confirm the use of "conform" or do you mean "confirm"?

Line 66 (pg 3): The word "Aurora" in the sentence " historical Anatolian aurora catalog (hAAc)" is not used in capital letter (A) due to the acronym "hAAc"? Please check this sentence and acronym throughout the paper. At this same line, it is important to mention the time period correspondent to the "medieval period" just after this sentence between parentheses.

Line 68 (pg 3): I suggest to replace "planetary climactic changes" for "Earth climactic changes".

Line 71/77/78 (pg 4): The acronym "hAAc" here appear as "hAAC". The author need to standardize this sentence and acronym. In the begging of this section it is interesting to show a map of Anatolian region, its borders at the current days with other countries (Today, which countries are in the Anatolian region? Turkey only our other countries), that is, the text in the lines 77/78 could be better explained in terms of the medieval period, and comparing that map with our current time map (actual geography) (Figure 1 could enclose two maps: the medieval period and the current time maps).

...

More minor comments will be posted soon.

---

## Referee Comment (RC2) · Anonymous Referee #2 · 28 Sep 2019

**Referee Report on MS angeo-2019-97 "Historical Aurora Borealis Observations in Anatolia during medieval period: Implications for the past solar activity" by N. Maden**

**General Comments**

This article has examined existing auroral catalogues, compiled auroral reports in Anatolia during the medieval period (apparently between 333 and 1143), and evaluated the "strength" of aurora with five criteria in Neuhäuser and Neuhäuser (2015). The compiled catalogue has been compared mainly with the Byzantine climatic records in Haldon et al. (2014) to discuss the solar-terrestrial relationship during this period. This manuscript is moderately interesting, as the Anatolian auroral records have not been comprehensively studied yet, and the author shows almost the opposite trend of solar activity around 774/775 against Neuhäuser and Neuhäuser (2015), using almost the same dataset and method with Neuhäuser and Neuhäuser (2015). However, this manuscript has to get its contents and novelty significantly improved for further considerations, as the auroral classification method is not very appropriate, the scientific discussions are not convincing enough, and the logic of his discussions on the climate change is extremely difficult to follow. Therefore, it is extremely important to improve the scientific novelty of this manuscript (see specific comments 1 and 2) for further considerations for publication in this journal.

**Specific Comments**

**1. Novelty of the Records**

The largest issue for this manuscript is its novelty, as the catalogued records are not new, classification methodology is not very appropriate, and scientific discussions are not quite sufficient. In order to improve the originality, the authors should consult not the existing catalogues but the original historical documents. This will let us improve accessibility to the original records improved and even potentially resolve apparent discrepancies in several records. The existing catalogues must not be misunderstood as the source documents, as done in Table 1. Showing an example of historical documents as a figure (see *e.g.*, Figures 1 – 2 of Kataoka et al., 2017; Figures 1 – 2 of Kataoka and Iwahashi, 2017) would be beneficial for the readership to understand what kind of historical records you are using in your article.

**2. "Strength of the Aurora"**

One of the scientific analyses in this article is the evaluation of "strength of the aurora" on the basis of criteria of Neuhäuser and Neuhäuser (2015). However, the author needs to explicitly clarify what the "strength of the aurora" means here. As long as reading Neuhäuser and Neuhäuser (2015), these criteria are not for strength but for likeliness. The strength of aurora is rather associated with the equatorward boundary of the aurora, as it has a good correlation with strength of magnetic storm (Yokoyama et al., 1998; Kataoka and Iwahashi, 2017). In this sense, stronger aurora will appear more southward and contradict the criteria for direction in Neuhäuser and Neuhäuser (2015). The author needs to revise and address the strength of aurora, citing Yokoyama et al. (1998) and Kataoka and Iwahashi (2017).

**3. The Validity of Criteria**

The author needs to seriously consider the validity of the criteria used in this manuscript and if they should be used in his manuscript. While the five criteria are based on (1) night-time (darkness, sunset, sunrise), (2) non-southern directions (northern, NE, NW, E-W, W-E), (3) color (red, reddish, fiery, bloody, green, black), (4) dynamics (fire, fiery), and (5) repetition, these criteria are unfortunately not consistent with observational evidence, as shown in Stephenson et al. (2019). I think the recent criticism makes good sense. Recent fact-based studies show that the equatorward boundaries of the aurora reach 25°, 24°, and 38° magnetic latitudes during the historical magnetic storms in 1770, 1859, and 1958 (Kimball, 1960; Kataoka and Iwahashi, 2017; Kataoka et al., 2019; Kataoka and Kazama, 2019). In the cases of such extreme space weather events, aurorae will be seen even southward from medieval Turkey (45 − 50.1° in magnetic latitude). It is also known that whitish pillar appears equatorward of the red glow during the strong magnetic storms, probably due to field-align currents carried by precipitating electrons (Kataoka et al., 2019). It is also not clear why fire or fiery means dynamics of aurora. The descriptions like "fire" more likely means auroral color and brightness (see Figure 1 of Kataoka and Kazama, 2019). The author needs to address these facts to evaluate validity of these criteria at the very least, if he strongly wishes to use these criteria in his manuscript. Otherwise, the author should not use these "criteria".

**4. Solar Activity around 774/775**

In scientific viewpoint, exploiting the discussions on the solar activity around 774/775 would benefit scientific community, as this is quite close to the cosmic ray event in 774/775 (*e.g.*, Miyake et al., 2012; Usoskin et al., 2013; Mekhaldi et al., 2015). The author seems to support the high solar activity (p.11; see also *e.g.*, Usoskin et al., 2013) with the reports and methods used in Neuhäuser and Neuhäuser (2015), whereas Neuhäuser and Neuhäuser (2015) suggested a solar minimum around 774. The author's result may be helpful to reconstruct the solar activity around 774/775, on which we have opposite reconstructions: low solar activity (Neuhäuser and Neuhäuser, 2015) and high solar activity (Usoskin et al., 2013; Stephenson et al., 2019). The author needs to clarify the scientific implications of his article for the solar activity around 774/775, evaluating the validity of the validity of Neuhäuser and Neuhäuser (2015).

**5. Chronological Coverage**

The author should define the survey object, namely the chronological extent of medieval Period and the geographical extent of Anatolia. Re chronological coverage, while the author's survey extent seems consistent with the former half of the Byzantine Empire (330 – 1453) in Haldon et al. (2014), the author should clarify why they stopped surveys in 1143.

**6. Definition of the Medieval Anatolia**

The definition of Anatolia is not clear as well. Geographically speaking, Constantinople is not in Anatolia but situated in the European side. The author needs to address why Asia Minor is exactly specified to be around current Ankara. It is also not very clear where is the border between Anatolia and Middle East. At least, it should not be the modern Turkish border. In my understanding, Edessa and Amida would be better located in the Middle East, rather than Anatolia.

**7. Relationship with Past Solar Activity**

The second conclusion in this manuscript states "In Anatolia and Middle East, there was a relatively high auroral activity during the years around 1100 is quite consistent with the naked-eye sunspot observations". However, the naked-eye sunspot observations are

mentioned only briefly in in the context of Medieval Maximum (p.12) and periodicity between 1095 and 1204 is usual (Vaquero and Trigo, 2012). Therefore, the author should compare these auroral records with the naked-eye sunspot observations. Moreover, the cycle length during the Medieval Maximum is probably shorter (~9 years) on the basis of $^{14}$C data (Miyahara et al., 2008) and their cycle reconstructions are shown in Kataoka et al. (2017). Hence the existing statement for solar cycle length needs to be revised, citing Miyahara et al. (2008) and Kataoka et al. (2017). This enhanced solar activity is also better illustrated, citing the earliest datable sunspot drawing and relevant Korean auroral records in 1128 (Willis and Stephenson, 2001; Willis and Davis, 2014), and contrasted with the Oort Minimum (Usoskin et al., 2007, 2017; see also Inceoglu et al., 2015).

**8. Relationship with Climatic Change**

While this manuscript is entitled as "Implications for the past solar activity" in its subtitle, the impacts on the climatic change has been emphasized in the manuscript (pp.13-14 and conclusions 5 – 6). However, the logic was extremely difficult to follow and the revision of humidity with auroral record has been applied without scientific explanations. The relationship between solar activity and climatic change in historical time span is not very clear (Vaquero and Trigo, 2012; Lockwood et al., 2017), while we know at least the lightning has correlation with solar rotation (Miyahara et al., 2017, 2018), and galactic cosmic ray fluence have some influence to snowball Earth (Kataoka et al., 2013, 2014) as well as explosive volcanic eruptions (Ebisuzaki et al., 2011). Therefore, the author is strongly recommended to separate their discussions for the climatic change to another article, indicating the solar-terrestrial relationship in short and very long time spans. This separation will make the logic in this manuscript more straightforward and improve its readability.

**9. Conclusions**

Accordingly, the conclusion needs to be modified. The second and third conclusions can be retained only if the author address naked-eye sunspot records appropriately. The fourth conclusion cannot co-exist with the third conclusion, as their coexistence make it unclear what was the main factor: solar activity or intensity of dipole moment and position of geomagnetic pole. The fifth and sixth conclusions should be separated to

another article, as well as the discussions on the climate change.

**Technical Corrections**

Technical corrections shown here are only those with relatively major importance. The author is strongly recommended to send this manuscript grammatical proofreading before resubmission, in order to improve the readability of this manuscript.

Line 28: For Chinese aurorae, cite Kataoka et al. (2017).

Line 27: For Japanese aurorae, cite Kataoka et al. (2017) and Kataoka et al. (2017). Remove Shiokawa et al. (2005), as this article is about modern instrumental observations.

Line 40-48: Remove this paragraph.

Line 109: The 502 August 22 event appears in the Zuqnin Chronicle too. Cite Hayakawa et al. (2017).

Line 131-155: The first observation in Zuqnin Chronicle should not be 772 but 771/772, namely somewhere between 771 October and 772 September, as the timing of harvest is not specified for a specific crop and there were multiple crops in Anatolia back then (Hayakawa et al., 2017).

Line 233-236: This statement should be brought somewhere before method, to clarify what the author surveyed.

Line 263-273: Separate this paragraph to another article.

Line 293-319: Separate these paragraphs to another article.

Line 324: "were thought" should be "thought"

Table 1: Remove it or replace it to a list of historical documents.

Table 2 and 4: The reference must be revised to the original historical documents.

Table 5: Remove it.

Figure 1: Remove the modern border and revise the location for Asia Minor.

Figure 2: Remove it.

Figure 3: Define the border of Anatolia and Middle East.

**References**

Ebisuzaki et al. (2011) Explosive volcanic eruptions triggered by cosmic rays: Volcano as a bubble chamber, *Gondwana Research*, **19**, 1054-1061.

Hayakawa, H., et al. (2017) The earliest drawings of 425 datable auroras and a two-tail comet from the Syriac Chronicle of Zūqnīn, *Publications of the Astronomical Society of Japan*, **69**, 17.

Inceoglu, F., et al. (2015) Grand solar minima and maxima deduced from 10Be and 14C: magnetic dynamo configuration and polarity reversal, *Astronomy & Astrophysics*, **577**, A20.

Kataoka, R., et al. (2013) Snowball Earth events driven by starbursts of the Milky Way Galaxy, *New Astronomy*, **21**, 50-62.

Kataoka, R., et al. (2014) The Nebula Winter: The united view of the snowball Earth, mass extinctions, and explosive evolution in the late Neoproterozoic and Cambrian periods, *Gondwana Research*, **25**, 1153-1163.

Kataoka, R., et al. (2017) Historical space weather monitoring of prolonged aurora activities in Japan and in China, *Space Weather*, **15**, 392-402.

Kataoka, R., Iwahashi, K. (2017) Inclined zenith aurora over Kyoto on 17 September 1770: Graphical evidence of extreme magnetic storm, *Space Weather*, **15**, 1314-1320.

Kataoka, R., et al. (2019) Fan-shaped aurora as seen from Japan during a great magnetic storm on February 11, 1958, *J. Space Weather Space Clim.*, **9**, A16.

Kataoka, R., Kazama, S. (2019) A watercolor painting of northern lights seen above Japan on 11 February 1958, *J. Space Weather Space Clim.*, **9**, A28.

Lockwood, M., et al. (2017) Frost fairs, sunspots and the Little Ice Age, *Astronomy & Geophysics*, **58**, 2.17–2.23.

Mekhaldi, F., et al. (2015) Multiradionuclide evidence for the solar origin of the cosmic-ray events of AD 774/5 and 993/4, Nature Communications, **6**, 8611.

Miyahara, H., Yokoyama, Y., Masuda, K. (2008) Possible link between multi-decadal climate cycles and periodic reversals of solar magnetic field polarity, *Earth and Planetary Science Letters*, **272**, 290–295

Miyahara, H., et al. (2017) Searching for the 27-day solar rotational cycle in lightning events recorded in old diaries in Kyoto from the 17th to 18th century, *Annales Geophysicae*, **35**, 1195-1200.

Miyahara, H., et al. (2018) Solar rotational cycle in lightning activity in Japan during the 18–19th centuries *Annales Geophysicae*, **36**, 633-640

Miyake, F., Nagaya, K., Masuda, K., Nakamura, T. (2012) A signature of cosmic-ray

increase in AD 774-775 from tree rings in Japan, *Nature*, **486**, 7402, 240-242

Neuhäuser, R., Neuhäuser, D. L. (2015) Solar activity around AD 775 from aurorae 472 and radiocarbon, *Astronomische Nachrichten*, **336**, 225–248

Stephenson, F. R., et al. (2019) Do the Chinese Astronomical Records Dated AD 776 January 12/13 Describe an Auroral Display or a Lunar Halo? A Critical Re-examination, *Solar Physics*, **294**, 36.

Usoskin, I. G. (2017) A history of solar activity over millennia, *Living Reviews in Solar Physics*, **14**, 3.

Usoskin, I. G., Solanki, S. K., Kovaltsov, G. A. (2007) Grand minima and maxima of solar activity: new observational constraints, *Astronomy and Astrophysics*, **471**, 301-309.

Usoskin, I. G., Kromer, B., Ludlow, F., Beer, J., Friedrich, M., Kovaltsov, G. A., Solanki, S. K., Wacker, L. (2013) The AD775 cosmic event revisited: the Sun is to blame, *Astronomy & Astrophysics*, **552**, L3.

Vaquero, J.M., Trigo, R.M. (2012) A Note on Solar Cycle Length during the Medieval Climate Anomaly, *Solar Physics*, **279**, 289-294.

Willis, D. M., Davis, C. J. (2015) Evidence for Recurrent Auroral Activity in the Twelfth and Seventeenth Centuries. In: Orchiston W., Green D., Strom R. (eds) *New Insights From Recent Studies in Historical Astronomy: Following in the Footsteps of F. Richard Stephenson*. Springer, Berlin.

Willis, D. M., Stephenson, F. R. (2001) Solar and auroral evidence for an intense recurrent geomagnetic storm during December in AD 1128, *Annales Geophysicae*, **19**, 289-302.

Yokoyama, N., Kamide, Y., Miyaoka, H. (1998) The size of the auroral belt during magnetic storms, *Annales Geophysicae*, **16**, 566-573.

---

## Author Comment (AC1) · 13 Nov 2019

Thank you for your constructive and helpful feedback, scholarly comments and timely processing of our submission. I have just revised the manuscript in view of the constructive and helpful editorial and reviewer comments as outlined in detail below and the paper is now ready to resubmit the journal of Annales Geophysicae (ANGEO) titled "Historical Aurora Borealis Observations in Anatolia during medieval period: Implications for the past solar activity". Please find our response to reviewer's comments step by step below.

Response to Anonymous Referee #1:

[Figure]

General Comments: In this paper, the author reports an overview of historical Aurora observations reports in Anatolia and Middle East regions in the medieval period based in historical texts, chronicles and aurora catalogs records. The paper tried to make a relationship between the auroral activity and the past solar activity, the past climatic changes, economy and society living in the remote time. My view on the paper is that though the discussions are interesting, but the paper do not bring clear new results and it was missing a interconnection between the historical facts, the beliefs of the ancient people and the new science that explain the Aurora phenomenon as a direct relation between the Sun events and the geomagnetic field, and also the current status of aurora observation in the north and south hemispheres. There are some scientific explanation that should be present in the paper and the cited time periods in the medieval era need to be more clear, explaining/discussing a little more some sentences and not just cite the previous papers or historical texts/chronicles. Due to these problems found in the paper I recommend to not accept the paper as it is presented. Reply: Thank you for your comment. The constructive comments by the reviewers are really appreciated.

Major Comments: 1) The title and other citation along the text do not include the approximate time period: e.g., in the title "Historical Aurora... during the medieval period...". I think that the approximate years or century could be cited. The correct location in time (type of calendar, year and/or century) should appear clearly in a historical text. Reply: The title of the manuscript is changed according to the reviewer comment.

2) In the Introduction section it was not defined/explained the Aurora phenomenon, neither the physical mechanisms responsible for the occurrence of this event (solar particle precipitation/solar wind, geomagnetic storms, and loss cone of particle perception/Earth magnetic field configuration, and the interaction of such charged/energetic particles with the neutral/ionized compounds of the upper atmosphere); for this kind of paper would be very interesting to show a couple examples of Auroras registered in the present time; it was missing the explanation that Auroras exist in both hemispheres

(North: Borealis Aurora; South: Austral Aurora), and the physical process/mechanism involved in the Aurora light emission is exactly the same, the magnetosphere and the Earth magnetic field configurations (approximately a magnetic dipole) and intensity have a very important role in the occurrence of Auroras (this was mentioned in some way, but not discussed and none map/cartoon was showed - for the present period). Auroras also can be observed in other planets besides Earth. In fact, I missed a deeper technical revision (including photos, a global map showing the Auroral regions in both north and south poles) in the introduction section. Reply: I would like to the Reviewer #1 for the encouraging and positive comments to improve the manuscript. The goal of this study is to compile a historical Anatolian aurora catalog (hAAC) during medieval period by scanning the available sources and catalogs in literature. The available catalogs described above present a number of records covering Europe, Japan, China, Russia and Middle East. There is no study dealing only with the historical aurora observations recorded in Anatolia. The catalog could be used to analyze the past solar activity and planetary climatic changes impacting on the economy and human events. This research may also contribute to the understanding of public perception of the historical auroras. Anatolia have not been studied until now with respect to meteorological and aurora observations. So, in the introduction section, it was not explained the physical mechanisms of the Aurora phenomenon. The Auroras in other planets seem entirely irrelevant.

3) In the Results and Discussion, I could not see discussion based on the summarized parts of historical records, and neither explanations of the relationship between Auroras and Climatic Changes/solar variability/society economy. These relations should be better discussed and explained in light of the current time when the climactic changes are discussed in a global scenario. Reply: Thanks to the Reviewer #1 for the constructive comments to improve the quality of the manuscript. In the "Results and Discussion" section, the aurora records and climate changes are discussed in detail in Line 304 to 322. Also, a discussion between Climate change and socio-economist growth is added to this section according to the reviewer comment.

4) In the Conclusions, it was stated that in the paper was established a relationship between the strong solar activity and auroral activity by integrating meteorological data (pg 15, lines 347-349). I could not see any meteorological data and evidence for such relationship along the paper or this was not stressed or adequate explained. On page 16, lines 363 and 364, the author suggested future investigations in order to establish a relationship between the solar variability and climatic changes. The current paper have the aim to obtain some relationship between these two phenomena, but it was not clear. Why the author do not use one of the suggestions (for example: "Medieval Climate Anomaly") and improve the current paper? This would be much more interesting than just make a revision on previous paper and texts from historical manuscripts, without a deep discussion. Reply: I would like to thank the Reviewer #1 for their thoughtful comments. The "meteorological data" is changed with the "historical-climatological data" throughout the manuscript. The sentence in the conclusion section is omitted.

Minor issues: Line 10/11 (pg 1): "...in order to understand the past solar activity and possible physical mechanism using historical texts, chronicles and other auroral records?". At the end of the reading it was not clear/understandable the physical mechanism beyond the auroras. Can the author improve the paper in order to satisfy this purpose? Reply: The sentence is revised according to the reviewer comments.

Line 14/15 (pg 1): "The data of the catalog strongly support that there is a considerable relationship between the aurora activity and past strong solar activity". Again, the paper did not clarify the above relationship. Reply: The sentence is revised according to the reviewer comments.

Line 16-18 (pg 1): "An unusually high auroral activity during the years around 1100... is quite consistent with the past solar variability, geomagnetic field intensity and planetary climatic changes". The text did not present clearly the relationship between the unusually high auroral activity around 1100 and the planetary climatic changes. Reply: I do not agree with the Reviewer #1. So, the sentence is not changed or deleted.

Line 58 (pg 3): "... maxima of auroral observations conform to the maxima in sunspot records..." . The author confirm the use of "conform" or do you mean "confirm"? Reply: The "conform" is changed with the "comply with".

Line 66 (pg 3): The word "Aurora" in the sentence " historical Anatolian aurora catalog (hAAc)" is not used in capital letter (A) due to the acronym "hAAc"? Please check this sentence and acronym throughout the paper. At this same line, it is important to mention the time period correspondent to the "medieval period" just after this sentence between parentheses. Reply: The "historical Anatolian aurora catalog" sentence and the "(hAAc)" acronym are checked throughout the paper.

Line 68 (pg 3): I suggest to replace "planetary climactic changes" for "Earth climactic changes". Reply: The "planetary climactic changes" terms are changed as "Earth climactic changes". Line 71/77/78 (pg 4): The acronym "hAAc" here appear as "hAAC". The author need to standardize this sentence and acronym. In the begging of this section it is interesting to show a map of Anatolian region, its borders at the current days with other countries (Today, which countries are in the Anatolian region? Turkey only our other countries), that is, the text in the lines 77/78 could be better explained in terms of the medieval period, and comparing that map with our current time map (actual geography) (Figure 1 could enclose two maps: the medieval period and the current time maps). Reply: The "hAAC" is revised as "hAAc". The Byzantine map for the medievep period is added to Figure 1.

Please also note the supplement to this comment:
https://www.ann-geophys-discuss.net/angeo-2019-97/angeo-2019-97-AC1-supplement.pdf

---

## Author Comment (AC2) · 13 Nov 2019

Thank you for your constructive and helpful feedback, scholarly comments and timely processing of our submission. I have just revised the manuscript in view of the constructive and helpful editorial and reviewer comments as outlined in detail below and the paper is now ready to resubmit the journal of Annales Geophysicae (ANGEO) titled "Historical Aurora Borealis Observations in Anatolia during medieval period: Implications for the past solar activity". Please find our response to reviewer's specific comments step by step below.

Response to Anonymous Referee #2:

General Comments This article has examined existing auroral catalogues, compiled auroral reports in Anatolia during the medieval period (apparently between 333 and 1143), and evaluated the "strength" of aurora with five criteria in NeuhaÌĹuser and NeuhaÌĹuser (2015). The compiled catalogue has been compared mainly with the Byzantine climatic records in Haldon et al. (2014) to discuss the solar-terrestrial relationship during this period. This manuscript is moderately interesting, as the Anatolian auroral records have not been comprehensively studied yet, and the author shows almost the opposite trend of solar activity around 774/775 against NeuhaÌĹuser and NeuhaÌĹuser (2015), using almost the same dataset and method with NeuhaÌĹuser and NeuhaÌĹuser (2015). However, this manuscript has to get its contents and novelty significantly improved for further considerations, as the auroral classification method is not very appropriate, the scientific discussions are not convincing enough, and the logic of his discussions on the climate change is extremely difficult to follow. Therefore, it is extremely important to improve the scientific novelty of this manuscript (see specific comments 1 and 2) for further considerations for publication in this journal. Reply: I would like to the Reviewer #2 for the encouraging and constructive comments to improve the quality of the manuscript.

Specific Comments 1. Novelty of the Records The largest issue for this manuscript is its novelty, as the catalogued records are not new, classification methodology is not very appropriate, and scientific discussions are not quite sufficient. In order to improve the originality, the authors should consult not the existing catalogues but the original historical documents. This will let us improve accessibility to the original records improved and even potentially resolve apparent discrepancies in several records. The existing catalogues must not be misunderstood as the source documents, as done in Table 1. Showing an example of historical documents as a figure (see e.g., Figures 1 – 2 of Kataoka et al., 2017; Figures 1 – 2 of Kataoka and Iwahashi, 2017) would be beneficial for the readership to understand what kind of historical records you are using in your article. Reply: Thanks to the reviewer #2 suggestions to improve the scientific content of the manuscript. The goal of this study is to compile a historical Anatolian

aurora catalog (hAAC) during medieval period by scanning the available sources and catalogs in literature. The available catalogs present a number of records covering Europe, Japan, China, Russia and Middle East. The aurora observations are collected from different historical text and available catalogs. For that reason, there is no figure like Figures 1 – 2 of Kataoka et al., 2017.

2. "Strength of the Aurora" One of the scientific analyses in this article is the evaluation of "strength of the aurora" on the basis of criteria of Neuhaĺ̇Luser and Neuhaĺ̇Luser (2015). However, the author needs to explicitly clarify what the "strength of the aurora" means here. As long as reading Neuhaĺ̇Luser and Neuhaĺ̇Luser (2015), these criteria are not for strength but for likeliness. The strength of aurora is rather associated with the equatorward boundary of the aurora, as it has a good correlation with strength of magnetic storm (Yokoyama et al., 1998; Kataoka and Iwahashi, 2017). In this sense, stronger aurora will appear more southward and contradict the criteria for direction in Neuhaĺ̇Luser and Neuhaĺ̇Luser (2015). The author needs to revise and address the strength of aurora, citing Yokoyama et al. (1998) and Kataoka and Iwahashi (2017). Reply: I would like to the Reviewer #2 for the encouraging and constructive comments to improve the quality of the manuscript. The study of Kataoka and Iwahashi (2017) and Yokoyama et al. (1998) is related to extention and auroral belt, respectively, not strength of Aurora. The sentence is revised as "One could decide whether an observation is strong aurorae by considering its color, brightness, dynamics, duration, geomagnetic latitude."

3. The Validity of Criteria The author needs to seriously consider the validity of the criteria used in this manuscript and if they should be used in his manuscript. While the five criteria are based on (1) night-time (darkness, sunset, sunrise), (2) non-southern directions (northern, NE, NW, E-W, W-E), (3) color (red, reddish, fiery, bloody, green, black), (4) dynamics (fire, fiery), and (5) repetition, these criteria are unfortunately not consistent with observational evidence, as shown in Stephenson et al. (2019). I think the recent criticism makes good sense. Recent fact-based studies show that the equatorward boundaries of the aurora reach 25°, 24°, and 38° magnetic latitudes during the historical magnetic storms in 1770, 1859, and 1958 (Kimball, 1960; Kataoka and Iwahashi, 2017; Kataoka et al., 2019; Kataoka and Kazama, 2019). In the cases of such extreme space weather events, aurorae will be seen even southward from medieval Turkey (45 − 50.1° in magnetic latitude). It is also known that whitish pillar appears equatorward of the red glow during the strong magnetic storms, probably due to field-align currents carried by precipitating electrons (Kataoka et al., 2019). It is also not clear why fire or fiery means dynamics of aurora. The descriptions like "fire" more likely means auroral color and brightness (see Figure 1 of Kataoka and Kazama, 2019). The author needs to address these facts to evaluate validity of these criteria at the very least, if he strongly wishes to use these criteria in his manuscript. Otherwise, the author should not use these "criteria". Reply: I would like to the Reviewer #2 for the encouraging and constructive comments to improve the quality of the manuscript. According to the study by Neuhäuser and Neuhäuser (2015), five criteria are implemented to perform the aurora catalogs as night-time (darkness, sunset, sunrise), non-southern directions (northern, NE, NW, E-W, W-E), color (red, reddish, fiery, bloody, green, black), dynamics (fire, fiery), and repetition. One could decide whether an observation is strong aurorae by considering its color, brightness, dynamics, duration, geomagnetic latitude. The observation is classified as potential (N=0), possible (N=1), very possible (N=2), probable (N=3), very probable (N=4), or certain (N=5) according to the criteria number (N) satisfied (Neuhäuser and Neuhäuser, 2015).

4. Solar Activity around 774/775 In scientific viewpoint, exploiting the discussions on the solar activity around 774/775 would benefit scientific community, as this is quite close to the cosmic ray event in 774/775 (e.g., Miyake et al., 2012; Usoskin et al., 2013; Mekhaldi et al., 2015). The author seems to support the high solar activity (p.11; see also e.g., Usoskin et al., 2013) with the reports and methods used in NeuhaİLuser and NeuhaİLuser (2015), whereas NeuhaİLuser and NeuhaİLuser (2015) suggested a solar minimum around 774. The author's result may be helpful to reconstruct the solar activity around 774/775, on which we have opposite reconstructions: low solar

activity (NeuhaÌLuser and NeuhaÌLuser, 2015) and high solar activity (Usoskin et al., 2013; Stephenson et al., 2019). The author needs to clarify the scientific implications of his article for the solar activity around 774/775, evaluating the validity of the validity of NeuhaÌLuser and NeuhaÌLuser (2015). Reply: I would like to the Reviewer #2 for the encouraging and constructive comments to improve the quality of the manuscript. Mekhaldi et al. (2015) indicated that these two extreme events (774/775) were five times greater than any other recorded solar storms with instruments. Their findings highlight the importance of studying the possibility of severe solar energetic particle events.

5. Chronological Coverage The author should define the survey object, namely the chronological extent of medieval Period and the geographical extent of Anatolia. Re chronological coverage, while the author's survey extent seems consistent with the former half of the Byzantine Empire (330 – 1453) in Haldon et al. (2014), the author should clarify why they stopped surveys in 1143. Reply: I would like to the Reviewer #2 for the encouraging and constructive comments to improve the quality of the manuscript. Figure 1 is revised according to the Reviewer #1 and #2. Any aurora observations could not be reached up to 1453.

6. Definition of the Medieval Anatolia The definition of Anatolia is not clear as well. Geographically speaking, Constantinople is not in Anatolia but situated in the European side. The author needs to address why Asia Minor is exactly specified to be around current Ankara. It is also not very clear where is the border between Anatolia and Middle East. At least, it should not be the modern Turkish border. In my understanding, Edessa and Amida would be better located in the Middle East, rather than Anatolia. Reply: I would like to the Reviewer #2 for the encouraging and constructive comments to improve the quality of the manuscript. Figure 1 is revised according to the Reviewer #1 and #2. The geographical border is changeable in the medieval period due to the wars between Turks and Byzantine Empire. So, the current border is displayed in this map. The places of the Constantinople, Amida, Edessa, Adana and Antioch are correct

geographically. The Asia Minor is other name of the Anatolia. So, the record belonging to Asia Minor (exact place not known) is located in the middle of the Anatolia capital of the Turkey.

7. Relationship with Past Solar Activity The second conclusion in this manuscript states "In Anatolia and Middle East, there was a relatively high auroral activity during the years around 1100 is quite consistent with the naked-eye sunspot observations". However, the naked-eye sunspot observations are mentioned only briefly in in the context of Medieval Maximum (p.12) and periodicity between 1095 and 1204 is usual (Vaquero and Trigo, 2012). Therefore, the author should compare these auroral records with the naked-eye sunspot observations. Moreover, the cycle length during the Medieval Maximum is probably shorter ($\sim$9 years) on the basis of 14C data (Miyahara et al., 2008) and their cycle reconstructions are shown in Kataoka et al. (2017). Hence the existing statement for solar cycle length needs to be revised, citing Miyahara et al. (2008) and Kataoka et al. (2017). This enhanced solar activity is also better illustrated, citing the earliest datable sunspot drawing and relevant Korean auroral records in 1128 (Willis and Stephenson, 2001; Willis and Davis, 2014), and contrasted with the Oort Minimum (Usoskin et al., 2007, 2017; see also Inceoglu et al., 2015). Reply: Thanks to the Reviewer #2 for the encouraging and constructive comments to improve the quality of the manuscript. The second conclusion is revised according to the comments. A detailed information about sun spot observations is added to the manuscript.

8. Relationship with Climatic Change While this manuscript is entitled as "Implications for the past solar activity" in its subtitle, the impacts on the climatic change has been emphasized in the manuscript (pp.13-14 and conclusions 5 – 6). However, the logic was extremely difficult to follow and the revision of humidity with auroral record has been applied without scientific explanations. The relationship between solar activity and climatic change in historical time span is not very clear (Vaquero and Trigo, 2012; Lockwood et al., 2017), while we know at least the lightning has correlation with solar rotation (Miyahara et al., 2017, 2018), and galactic cosmic ray fluence have some

influence to snowball Earth (Kataoka et al., 2013, 2014) as well as explosive volcanic eruptions (Ebisuzaki et al., 2011). Therefore, the author is strongly recommended to separate their discussions for the climatic change to another article, indicating the solar-terrestrial relationship in short and very long time spans. This separation will make the logic in this manuscript more straightforward and improve its readability. Reply: I would like to the Reviewer #2 for the encouraging and constructive comments to improve the quality of the manuscript. This study could be significant constraints for exploration of solar activity on Earth's atmosphere and climate during the historical periods previously proved by Bard and Frank (2006). According to the Bard and Frank (2006) solar fluctuations caused climatic changes called Medieval Warm Period (900–1400). The Maunder Minimum (1645-1715) which delineates the coldest part of the Little Ice Age (Eddy, 1976) is depicted by a solar activity reduction, as well as a sunspots scarcity. The Medieval Climate Anomaly characterizing by warmer and drier climate conditions generally related to reasonably prolonged solar activity during the 12th and 13th centuries (Jirikowic and Damon, 1994).

9. Conclusions Accordingly, the conclusion needs to be modified. The second and third conclusions can be retained only if the author address naked-eye sunspot records appropriately. The fourth conclusion cannot co-exist with the third conclusion, as their co-existence make it unclear what was the main factor: solar activity or intensity of dipole moment and position of geomagnetic pole. The fifth and sixth conclusions should be separated to another article, as well as the discussions on the climate change. Reply: I do not agree with the Reviewer #2. So, it is not suitable for removing these conclusions from the manuscript.

Technical Corrections Technical corrections shown here are only those with relatively major importance. The author is strongly recommended to send this manuscript grammatical proofreading before resubmission, in order to improve the readability of this manuscript. Line 28: For Chinese aurorae, cite Kataoka et al. (2017). Reply: Ok

Line 27: For Japanese aurorae, cite Kataoka et al. (2017) and Kataoka et al. (2017).

Remove Shiokawa et al. (2005), as this article is about modern instrumental observations. Reply: OK

Line 40-48: Remove this paragraph. Reply: OK Line 109: The 502 August 22 event appears in the Zuqnin Chronicle too. Cite Hayakawa et al. (2017). Reply: OK

Line 131-155: The first observation in Zuqnin Chronicle should not be 772 but 771/772, namely somewhere between 771 October and 772 September, as the timing of harvest is not specified for a specific crop and there were multiple crops in Anatolia back then (Hayakawa et al., 2017). Reply: Revised

Line 233-236: This statement should be brought somewhere before method, to clarify what the author surveyed. Reply: The statement is added to the "Introduction" section.

Line 263-273: Separate this paragraph to another article. Reply: Revised

Line 293-319: Separate these paragraphs to another article. Reply: Revised

Line 324: "were thought" should be "thought" Reply: Revised

Table 1: Remove it or replace it to a list of historical documents. Reply: Revised

Table 2 and 4: The reference must be revised to the original historical documents. Reply: The Reference list is revised.

Table 5: Remove it. Reply: This Figure is important to understand the climate change in Anatolia. So, it should not be removed from the manuscript.

Figure 1: Remove the modern border and revise the location for Asia Minor. Reply: Figure 1 is revised

Figure 2: Remove it. Reply: Again, Figure 2 is important to understand the climate change in Anatolia. So, it should not be removed from the manuscript.

Figure 3: Define the border of Anatolia and Middle East. Reply: Thank you for your comment. The constructive comments by the reviewers are really appreciated. Figure

3 is not a map, but a histogram plot. So, there is no border. The aurora observations are divided in to two panels for Anatolia and Middle east region.

We thank to you and the Reviewer #1 for the constructive and helpful comments.

Sincerely, Dr. Nafiz MADEN

Please also note the supplement to this comment:
https://www.ann-geophys-discuss.net/angeo-2019-97/angeo-2019-97-AC2-supplement.pdf

---

## Referee Report (RR1)

**Second Referee Report on MS angeo-2019-97 "Historical Aurora Borealis Observations in Anatolia during medieval period: Implications for the past solar activity" by N. Maden**

**General Comments**

With regret, I found that the manuscript is virtually the same with the previous revision. I seriously suspect if the author has really revised the manuscript after receiving two referee reports. Even in the response, the author has failed addressing most of my previous comments and clarifying the novelty of this manuscript. The author has almost explicitly admitted that he has not consulted the original historical documents for his survey. He has failed to explain the strength of aurora and has not done anything more than repeating what Neuhäuser and Neuhäuser (2015) have written. More seriously, the revised manuscript does not involve revised phrases, while the author claimed to have done. Overall, I cannot consider this manuscript has been improved through the referee process. While I am extremely reluctant to repeat my previous comments, I have had to do that, as my previous comments have not been addressed appropriately. Unless otherwise the author has mistakenly uploaded a wrong file, I would recommend rejection of this manuscript.

**Specific Comments**

**1. Serious Discrepancies between the Revision and Response**

This revision made me seriously doubt if the author has indeed revised the manuscript after receiving two referee reports. Indeed, I found various discrepancies between the revision and the response. For example, reg. my major comment 2, the author stated "The sentence is revised as "One could decide whether an observation is strong aurorae by considering its color, brightness, dynamics, duration, geomagnetic latitude"", whereas this kind of phrase was not found in the main manuscript. Likewise, reg. my major comment 7, the author stated "A detailed information about sun spot observations is added to the manuscript", while this kind of statement could not be found in the revision, either. The author has agreed to cite several overlooked references (e.g., Kataoka et al., 2017; Kataoka and Iwahashi, 2017, etc.), whereas they are not found in the revision. After all, the revision and response are seriously inconsistent with each other and made me seriously doubt if the author has indeed made any revision upon this

manuscript.

As the manuscript is – at least apparently – not revised at all, I do not think it would be meaningful to comment anything more than what I commented in the previous review. My following comments are not my comments on the revision, but my answers against what the author has written in his response letter.

**2. Novelty of the Records**

As I commented before, the largest issue for this manuscript is its novelty. What the author has done in this manuscript is to simply recompile the Anatolian auroral reports from the existing catalogs (not from the original historical documents!). Therefore, these presented results are unfortunately not new. The scientific method is currently no more than a repetition of Neuhäuser and Neuhäuser (2015), while the author's outcome for the solar activity around 774/775 contradict what Neuhäuser and Neuhäuser (2015) have concluded. In this case, the only potential novelty of this manuscript is – at best – the emphasis of the high solar activity around 774/775. As long as I understand, "ANGEO publishes *original* articles and short communications (letters) on research of the Sun–Earth system...". Therefore, the originality of this manuscript is crucially important to let this manuscript get subjected to further considerations.

**3. "Strength of the Aurora"**

The author must read Neuhäuser and Neuhäuser (2015) more carefully. Neuhäuser and Neuhäuser (2015) have explicitly stated "we establish five criteria for the *likeliness* of the event to be an aurora which are selected to distinguish from the other effects" in page 230. As the author has cited "The observation is classified as potential (N=0), possible (N=1), very possible (N=2), N aurora is probable (N=3), very probable (N=4), or certain (N=5) according to the criteria number (N) satisfied". This is not about strength but about likeliness. As the equatorward extension of auroral oval has good correlation with "strength" of magnetic storm (Yokoyama et al., 1998), the "strength" would be better understood with the equatorward extension of auroral oval. Therefore, repeating an excerpt from Neuhäuser and Neuhäuser (2015) does not make any good sense here.

**4. The Validity of Criteria**

More seriously, the author has entirely failed to address the scientific concern for the validity of Neuhäusers' criteria, only repeating what Neuhäusers described. As I commented previously, their criteria have been seriously doubted with counter-examples (Stephenson et al., 2019). The fact-based studies show that the equatorward boundaries of the aurora reach 25°, 24°, and 38° magnetic latitudes during the historical magnetic storms in 1770, 1859, and 1958 (Kimball, 1960; Kataoka and Iwahashi, 2017; Kataoka et al., 2019; Kataoka and Kazama, 2019). In the cases of such extreme space weather events, aurorae will be seen even southward from medieval Turkey (45 – 50.1° in magnetic latitude). It is also known that whitish pillar appears equatorward of the red glow during the strong magnetic storms, probably due to field-align currents carried by precipitating electrons (Kataoka et al., 2019). It is also not clear why fire or fiery means dynamics of aurora. The descriptions like "fire" more likely means auroral color and brightness (see Figure 1 of Kataoka and Kazama, 2019). The author needs to address these facts to evaluate validity of these criteria at the very least, if he strongly wishes to use these criteria in his manuscript. Otherwise, the author must not use these "criteria".

**5. Solar Activity around 774/775**

While I appreciate scientific contribution by Mekhaldi et al. (2015), Neuhäuser and Neuhäuser (2015) have claimed "they [their auroral records] cannot support a hypothetical solar super-flare" in page 236, for example. This is almost in an opposite spectrum against Mekhaldi et al. (2015). The author needs to clarify what he can say from Anatolian records for such scientific conflict.

**6. Chronological Coverage**

Why "Any aurora observations could not be reached up to 1453"? That must be scientifically explained.

**7. Definition of the Medieval Anatolia**

As the medieval border is changeable, it is even more misleading to plot the present Turkish border. The border should be removed from the map. As the Turkish came into Anatolia only after the battle of Malazgirt in 1071 (e.g., Barber, 2012), it is misleading

to consider the Byzantine Anatolia as something equivalent with the territory of modern Turkey. Constantinople is situated on the European side and outside of Anatolia. In Turkish, it is geographically categorized as "Rumelia". Eddesa and Amida are situated in Mesopotamia. Therefore, they are not in Anatolia either.

**8. Relationship with Climatic Change**

As I commented previously, the logic was extremely difficult to follow and the revision of humidity with auroral record has been applied without scientific explanations. The author needs to seriously note that the relationship between solar activity and climatic change in historical time span is not very clear (Vaquero and Trigo, 2012; Lockwood et al., 2017). Lockwood et al. (2017) have especially clarified how misleading to explain the Little Ice Age with the Maunder Minimum. They have casted a caveat "The association of the solar Maunder minimum and the Little Ice Age is also not supported by proper inspection and ignores the role of other factors such as volcanoes" in page 2.23 for example. This made me strongly doubt the validity of the author's discussion for climatological impact. This manuscript cannot be published, unless otherwise the author removes their speculation about the climatic impact.

**9. Conclusion**

Accordingly, the fifth and sixth conclusions must be removed, as well as the discussions on the climate change. In the same time, the author needs to clarify which made aurora visible in Anatolia so frequently in the Byzantine period: solar activity or intensity of dipole moment and position of geomagnetic pole.

---

## Referee Report (RR2)

**Third Referee Report on MS angeo-2019-97 "Historical Aurora Borealis Observations in Anatolia during medieval period: Implications for the past solar activity" by N. Maden**

**General Comments**

With regret, I found that the author has failed to address most of my previous comments or clarify the novelty of this manuscript, while the author's version is slightly better than the virtually unchanged previous version. The author has almost explicitly admitted that he has not consulted the original historical documents for his survey. He has failed to explain the strength of aurora and has not done anything more than repeating what Neuhäuser and Neuhäuser (2015) have written, while their criteria themselves contradict the actual observational evidence (see *e.g.*, Stephenson et al., 2019). With great respect, I have to comment that applying dubious criteria to non-original records would not guarantee a novelty for an academic article, at least in *Annales Geophysicae*. Overall, I cannot recommend its publication in *Annales Geophysicae*, unless otherwise the author seriously revises this manuscript from its basis.

**Specific Comments**

**1. Novelty of the Records**

As I commented before, the largest issue for this manuscript is its novelty. What the author has done in this manuscript is to simply recompile the Anatolian auroral reports from the existing catalogs (not from the original historical documents!). Therefore, these presented results are unfortunately not new. The scientific method is currently no more than a repetition of Neuhäuser and Neuhäuser (2015), while the author's outcome for the solar activity around 774/775 contradict what Neuhäuser and Neuhäuser (2015) have concluded. In this case, the only potential novelty of this manuscript is – at best – the emphasis of the high solar activity around 774/775. As long as I understand, "ANGEO publishes *original* articles and short communications (letters) on research of the Sun–Earth system...". Therefore, the originality of this manuscript is crucially important to let this manuscript get subjected to further considerations.

**2. "Strength of the Aurora"**

The author must read Neuhäuser and Neuhäuser (2015) more carefully. Neuhäuser and Neuhäuser (2015) have explicitly stated "we establish five criteria for the *likeliness* of the event to be an aurora which are selected to distinguish from the other effects" in page 230. As the author has cited "The observation is classified as potential (N=0), possible (N=1), very possible (N=2), N aurora is probable (N=3), very probable (N=4), or certain (N=5) according to the criteria number (N) satisfied". This is not about strength but about likeliness. As the equatorward extension of auroral oval has good correlation with "strength" of magnetic storm (Yokoyama et al., 1998), the "strength" would be better understood with the equatorward extension of auroral oval. Therefore, repeating an excerpt from Neuhäuser and Neuhäuser (2015) does not make any good sense here.

**3. The Validity of Criteria**

Even more seriously, the author has entirely failed to address the scientific concern for the validity of Neuhäusers' criteria, only repeating what Neuhäusers described. As I commented previously, their criteria have been seriously doubted with counter-examples (Stephenson et al., 2019). The fact-based studies show that the equatorward boundaries of the aurora reach 25°, 24°, and 38° magnetic latitudes during the historical magnetic storms in 1770, 1859, and 1958 (Kimball, 1960; Kataoka and Iwahashi, 2017; Kataoka et al., 2019; Kataoka and Kazama, 2019). In the cases of such extreme space weather events, aurorae will be seen even southward from medieval Turkey (45 – 50.1° in magnetic latitude). It is also known that whitish pillar appears equatorward of the red glow during the strong magnetic storms, probably due to field-align currents carried by precipitating electrons (Kataoka et al., 2019). It is also not clear why fire or fiery means dynamics of aurora. The descriptions like "fire" more likely means auroral color and brightness (see Figure 1 of Kataoka and Kazama, 2019). The author needs to address these facts to evaluate validity of these criteria at the very least, if he strongly wishes to use these criteria in his manuscript. Otherwise, the author must not use these "criteria".

**4. Solar Activity around 774/775**

While I appreciate scientific contribution by Mekhaldi et al. (2015) on the extreme solar storm in 774/775, Neuhäuser and Neuhäuser (2015) have claimed "they [their auroral

records] cannot support a hypothetical solar super-flare" in page 236, for example. This is almost in an opposite spectrum against Mekhaldi et al. (2015). The author needs to clarify what he can say from Anatolian records for such scientific conflict.

**5. Chronological Coverage**

Why "Any aurora observations could not be reached up to 1453"? That must be scientifically explained. Moreover, the title of "medieval Anatolian" should be revised to "Byzantine" or "Byzantine Anatolian", given what the author surveyed.

**6. Definition of the Medieval Anatolia**

The author needs to see Figure 1 a little more carefully. Edessa (and probably Amida too) is/are situated outside of the Byzantine territory. More seriously, this figure explicitly shows that Constantinople is situated not in Asia Minor but in "Macedonia", while majority of the records in the author's catalog are derived from Constantinople. Therefore, Figure 1 shows that they are not in Anatolia either.

**7. Relationship with Past Solar Activity**

While I commented on this aspect, the author just cited Willis and Stephenson (2001) without enough explanation. I cannot consider the author's addition as a "detailed explanation". Therefore, I have to repeat what I have written previously. The second conclusion in this manuscript states "In Anatolia and Middle East, there was a relatively high auroral activity during the years around 1100 is quite consistent with the naked-eye sunspot observations". However, the naked-eye sunspot observations are mentioned only briefly in in the context of Medieval Maximum (p.12) and periodicity between 1095 and 1204 is usual (Vaquero and Trigo, 2012). Therefore, the author should compare these auroral records with the naked-eye sunspot observations. Moreover, the cycle length during the Medieval Maximum is probably shorter (~9 years) on the basis of $^{14}$C data (Miyahara et al., 2008) and their cycle reconstructions are shown in Kataoka et al. (2017). Hence the existing statement for solar cycle length needs to be revised, citing Miyahara et al. (2008) and Kataoka et al. (2017). This enhanced solar activity is also better illustrated, citing the earliest datable sunspot drawing and relevant Korean auroral records in 1128 (Willis and Stephenson, 2001; Willis and Davis, 2014), and contrasted with the Oort Minimum (Usoskin et al., 2007, 2017; see also Inceoglu et al.,

2015).

**8. Relationship with Climatic Change**

As I commented previously, the logic was extremely difficult to follow and the revision of humidity with auroral record has been applied without scientific explanations. The author needs to seriously note that the relationship between solar activity and climatic change in historical time span is not very clear (Vaquero and Trigo, 2012; Lockwood et al., 2017). Lockwood et al. (2017) have especially clarified how misleading to explain the Little Ice Age with the Maunder Minimum. They have casted a caveat "The association of the solar Maunder minimum and the Little Ice Age is also not supported by proper inspection and ignores the role of other factors such as volcanoes" in page 2.23 for example. This made me strongly doubt the validity of the author's discussion for climatological impact. This manuscript cannot be published, unless otherwise the author removes their speculation about the climatic impact.

**9. Conclusion**

Accordingly, the fifth and sixth conclusions must be removed, as well as the discussions on the climate change. In the same time, the author needs to clarify which made aurora visible in Anatolia so frequently in the Byzantine period: solar activity or intensity of dipole moment and position of geomagnetic pole.

---

## Referee Report (RR3)

Review Manuscript Number: angeo-2019-97

Title: "Historical Aurora Borealis Observations in Anatolia during the medieval period (AD 1-1453): Implications for the past solar activity"

Author: Nafiz Maden

**General Comments:**

In this paper, the author reports an overview of historical Aurora observations reports in Anatolia and Middle East regions in the medieval period based in historical texts, chronicles and aurora catalogs records. The paper presented a relationship between the auroral activity and the past solar activity, the past climatic changes, economy and society living in the remote time.

My view on the paper is that the discussions are interesting, and more discussions and clarifications were made in the revised version and the full paper desire publication in the ANGEO.

**Major Comments:**

All the major comments in the previous version was answered in a proper way, and at this point I do not have any major recommendation.

**Minor issues:**

Line 1 (pg 1): In the title would not more correct to add the word "the" before "medieval period"? and also one space after "AD …"?

Line 73 (pg 4): In the title of the section 2, I think that the word "Catalog" for the hAAc acronyms should be with "c" instead of "C".

Line 231 (pg 10): I suggest changing the sentence "23 different historical aurora records…" for "Twenty-three different aurora records…".

Line 357: In the Conclusions section, add the words "the" and "field" just after in the following sentence (check the correct language in this case): "… important information on variations in the geomagnetic field and auroral activity…"

The last suggestion is to enumerate the pages in the button right side or according to the journal standard.

Sincerely yours,

**Dr. José Valentin Bageston**
Reviewer # 1

---

## Referee Report (RR4)

**Third Referee Report on MS angeo-2019-97 "Historical Aurora Borealis Observations in Anatolia during medieval period: Implications for the past solar activity" by N. Maden**

**General Comments**

Apparently, the author has slightly improved this manuscript but the revision seems only stopgap and cannot resolve the problems at its roof. Therefore, I have addressed my concerns as positively as possible and tried to show how to revise this manuscript more explicitly. The novelty issue should be resolved pushing scientific implications for the medieval grand maximum and the extreme solar particle storm in 774.775, as the author does not have any novelty in the source records. The usage of "Anatolia" is highly problematic, as Constantinople, providing more than half of involved reports, is situated in the European side. The author needs to be more explicit about the obtained implication for the solar activity around the extreme solar particle storm in 774/775. The climatological discussions must be removed as the current evidence does not satisfy scientific threshold and will reduce the value of this manuscript. Overall, these comments are minimal requirements for publication in the Annales Geophysicae, which "publishes original articles and short communications (letters) on research of the Sun–Earth system...". The author is dully requested to address these comments appropriately and improve his scientific discussions and English grammar.

**Specific Comments**

**1. Novelty of the Records**

Unfortunately, compiling local auroral reports from existing catalogs does not guarantee novelty. This is especially the case, as the author explicitly admitted that he has not consulted the original historical documents and declined to provide example images of the original historical documents. The readers *would* have found its novelty, if the author *extracted* auroral records not from existing catalogs but from original historical documents.

Even more seriously, more than half of the auroral reports in this catalog (9 out of 14) are derived not from Anatolia but from Constantinople. As Constantinople is situated in the European side ("Macedonia" in the Byzantine Epoch or "Rumelia" in the Ottoman

Epoch), they are not classified as "historical Anatolian Aurora". This is explicitly shown in Figure 1. Therefore, this catalog must drop these 9 records, in order to let this manuscript be a "historical Anatolian Aurora catalog".

Therefore, the author needs to show its novelty in his scientific discussions, as these data are not new and more than half of them are not from Anatolia.

**2. "Strength of the Aurora" and their Validity**

This has been much improved, removing misleading usage of likeliness evaluations. However, the narrative in L61-70 is then too long. Given the author's existing discussions, this lengthy phrase ("Neuhäuser and Neuhäuser (2015) are implemented ... cannot be classified as extreme events associated with extreme magnetic storms.") should be revised and connected more to their scientific discussions as follows:

"Recently, such candidate records of mid-latitude aurorae have been intensively investigated (*e.g.*, Usoskin et al., 2013; Stephenson, 2015), due to the discovery of footprints of an extreme solar particle storm in the cosmogenic isotopes around 774/775 (Miyake *et al*., 2012; Usoskin *et al*., 2013; Mekhaldi *et al*., 2015). While Neuhäuser and Neuhäuser (2015) suggsted five likeliness "criteria" and rejected most of the candidate aurorae around this event. However, these criteria actually contradicted auroral behaviour during the extreme space weather events (Kimball, 1960; Kataoka and Iwahashi, 2017; Kataoka *et al*., 2019; Kataoka and Kazama, 2019). Indeed, Stephenson *et al*. (2019) rejected these criteria and their analyses on the basis of multiple counter-examples during the extreme space weather events and confirmed an enhanced solar activity around this epoch. Their conclusion is consistent with the isotope evidence for the extreme solar particle storm such as the detected ratio of Be10 and Cl36 (Mekhaldi *et al*., 2015), latitudinal concentration of C14 concentration (Uusitalo *et al*., 2018), and coincidental spikes of the multiple cosmogenic isotopes in both hemispheres (Büngten *et al*., 2018)."

**3. Solar Activity around 774/775**

In order to push their scientific novelty, the author needs to expand this section, rather than dropping it. Extending what the author has written, I would suggest writing as

follows, on the basis of what the author has claimed.

"The low-latitude aurorae of 772-773 are interesting, as being very close to the extreme solar event of 774/775 (Miyake et al., 2012; Usoskin et al., 2013; Mekhldi et al., 2015). These low-latitude aurorae are quite close from the extreme solar particle storm in 774/775 and support not the solar minimum (Neuhäuser and Neuhäuser, 2015) but high solar activity back then (Usoskin *et al*., 2013; Mekhaldi *et al*., 2015; Stephenson *et al*., 2019)."

**4. Chronological Coverage**

To say "1453 is considered the end of the medieval period by historians", the authors must provide evidence. This is the end of Byzantine Empire, not the medieval epoch. I do not think the Ottoman conquest of the Constantinople is a benchmark of the medieval epoch. After all, it is not "medieval Anatolia" but "Byzantine Anatolia" that the author surveyed.

**5. Definition of the Medieval Anatolia**

The revision of Figure 1 can be mistaken as concealment, as Constantinople is anyways situated not in the Anatolian side but in the European side. The existing title with "Anatolia" is anyways highly misleading. If the author wishes to keep this title, the author must drop 9 records from Constantinople. The territory of Anatolia and modern Republic of Turkey is not the same.

**6. Relationship with Past Solar Activity and Climate Change**

As I commented before, the author must not mix up the solar activity and the terrestrial climate changes (see Vaquero and Trigo, 2012; Lockwood et al., 2017). As the author does not have a clear tie between the medieval solar maximum and medieval warm period, the author needs to discard almost everything between P12L269 and P14L309: "*This study could also be significant constraints for exploration of solar activity on Earth's atmosphere and climate during the historical periods previously proved by Bard and Frank (2006). ... An important increase in agricultural production and population seems to have occurred in Anatolia after the year of 1100*". If the author wishes to claim this relationship, he needs more supporting evidence and write another

article.

Then, the author needs to rewrite his discussion on the medieval grand maximum focusing not on the periodicity but on the amplitude of solar cycles. I would suggest writing as follows.

"Vaquero and Trigo (2012) stated the period from 1095 to 1204 as an average solar cycle length, whereas this needs to be carefully compared with the reconstructed solar cycles on the basis of cosmogenic isotopes (Miyahara *et al.*, 2008; Kataoka *et al.*, 2017). Nevertheless, this period is characterised with numerous records of sunspots and aurorae shown in Vaquero and Vazquez (2009) and supported by Anatolian reports compiled in this article. This is highly consistent with an appearance of a gigantic sunspot in 1128 that caused a serious geomagnetic storm (Willis and Stephenson, 2001) and contrasts well with the Oort Minimum (Usoskin *et al.*, 2007, 2017; see also Inceoglu *et al.*, 2015). Indeed, Bekli et al. (2017) demonstrated that the naked-eye sun spot observations from 974 to 1278 and aurora records from 965 to 1273 show multiple unusual peaks related to the high solar activitiy at latitudes below 45° by using Chinese and Korean historical sources."

**9. Conclusion**

I appreciate that the author compiled auroral reports in Anatolia and Balkan Peninsula (Constantinople) from existing catalogs and compared them with other scientific results. However, unfortunately, what the author did in the climatological context does not satify scientific threshold and needs much more scientific supports. This can be done only writing another article for that issue. Therefore, the fifth and sixth conclusions must be removed, as well as the discussions on the climate change. Instead, the author should add their finding on the high solar activity around the extreme solar particle storm in 774/775 in the conclusion.

**Minor Comments**

P1L14-17: "High Aurora activity during the years around 1100 in Anatolia and Middle East is quite consistent with the past solar variability and planetary climatic changes drastically impacting on the economy and human events." => "High auroral activity

around the extreme solar particle storm in 774/775 and the medieval grand maximum in 1100s in Anatolia and Middle East is quite consistent with the past solar variability reported in other scientific literature"

P7L145: Neuhäuser and Neuhäuser (2015) did not do anything more than Harrak (1999) for Zuqnin Chronicle. Just cite Harrak (1999). These Zuqnin records have been intensively analysed in Hayakawa et al. (2017). Cite it here and Table 2 #11.

P8L187: Cite references for definition of the Armenian years.

P14L313: "Medieval grand maximum" should not be mixed up with "Medieval Climate Anomaly".

---

## Author Response (AR3)

Dear Dr. Igo Paulino,
Topical Editor
Annales Geophysicae (ANGEO)

**Ref**         **:** angeo-2019-97
**Title**       **:** Historical Aurora Borealis Observations in Anatolia during medieval
                period: Implications for the past solar activity
**Journal**     **:** Annales Geophysicae (ANGEO)

  Thank you for your constructive and helpful feedback, scholarly comments and timely processing of our submission. I have just revised the manuscript in view of the constructive and helpful editorial and reviewer comments as outlined in detail below and the paper is now ready to resubmit the journal of Annales Geophysicae (ANGEO) titled "Historical Aurora Borealis Observations in Anatolia during medieval period: Implications for the past solar activity". Please find our response (in red) to reviewer's specific comments (in black) step by step below.

  I would like to thank the reviewers for their thoughtful comments. Responses to comments are presented in the following pages along with explanations.

  Thanks again and looking forward to hearing from you soon.

  Best regards,
  **Dr. Nafiz MADEN**
  Corresponding author

**Detailed Response to Reviewers**

***Response to comments from Anonymous Referee #1:***

***General Comments:***

With regret, I found that the author has failed to address most of my previous comments or clarify the novelty of this manuscript, while the author's version is slightly better than the virtually unchanged previous version. The author has almost explicitly admitted that he has not consulted the original historical documents for his survey. He has failed to explain the strength of aurora and has not done anything more than repeating what Neuhäuser and Neuhäuser (2015) have written, while their criteria themselves contradict the actual observational evidence (see e.g., Stephenson et al., 2019). With great respect, I have to comment that applying dubious criteria to non-original records would not guarantee a novelty for an academic article, at least in Annales Geophysicae. Overall, I cannot recommend its publication in Annales Geophysicae, unless otherwise the author seriously revises this manuscript from its basis.
**Reply:** *I would like to thank the Reviewer #1.*

***Specific Comments***
***1. Novelty of the Records***

As I commented before, the largest issue for this manuscript is its novelty. What the author has done in this manuscript is to simply recompile the Anatolian auroral reports from the existing catalogs (not from the original historical documents!). Therefore, these presented results are unfortunately not new. The scientific method is currently no more than a repetition of Neuhäuser and Neuhäuser (2015), while the author's outcome for the solar activity around 774/775 contradict what Neuhäuser and Neuhäuser (2015) have concluded. In this case, the only potential novelty of this manuscript is – at best – the emphasis of the high solar activity around 774/775. As long as I understand, "ANGEO publishes original articles and short communications (letters) on research of the Sun–Earth system...". Therefore, the originality of this manuscript is crucially important to let this manuscript get subjected to further considerations.
**Reply:** *Thank you for encouraging comments to improve the manuscript. The novelty of this manuscript is given below:*
*There is no study dealing only with the historical aurora observations recorded in Anatolia. Anatolia have not been studied until now with respect to historical-climatological data and aurora observations. The goal of this study is to compile a historical Anatolian Aurora catalog (hAAC) during the medieval period by scanning the available sources and catalogs in literature.*

***2. "Strength of the Aurora"***

The author must read Neuhäuser and Neuhäuser (2015) more carefully. Neuhäuser and Neuhäuser (2015) have explicitly stated "we establish five criteria for the likeliness of the event to be an aurora which are selected to distinguish from the other effects" in page 230. As the author has cited "The observation is classified as potential (N=0), possible (N=1), very possible (N=2), N aurora is probable (N=3), very probable (N=4), or certain (N=5) according to the criteria number (N) satisfied". This is not about strength but about likeliness. As the equatorward extension of auroral oval has good correlation with "strength" of magnetic storm (Yokoyama et al., 1998), the "strength" would be better understood with the equatorward extension of auroral oval. Therefore, repeating an excerpt from Neuhäuser and Neuhäuser (2015) does not make any good sense here.

*Reply: I would like to thank the Reviewer #1 for encouraging comments to improve this study. The paragraph is revised according to the recent study performed by Stephenson et al., 2019.*

**3. The Validity of Criteria**

Even more seriously, the author has entirely failed to address the scientific concern for the validity of Neuhäusers' criteria, only repeating what Neuhäusers described. As I commented previously, their criteria have been seriously doubted with counter-examples (Stephenson et al., 2019). The fact-based studies show that the equatorward boundaries of the aurora reach 25°, 24°, and 38° magnetic latitudes during the historical magnetic storms in 1770, 1859, and 1958 (Kimball, 1960; Kataoka and Iwahashi, 2017; Kataoka et al., 2019; Kataoka and Kazama, 2019). In the cases of such extreme space weather events, aurorae will be seen even southward from medieval Turkey (45 – 50.1° in magnetic latitude). It is also known that whitish pillar appears equatorward of the red glow during the strong magnetic storms, probably due to field-align currents carried by precipitating electrons (Kataoka et al., 2019). It is also not clear why fire or fiery means dynamics of aurora. The descriptions like "fire" more likely means auroral color and brightness (see Figure 1 of Kataoka and Kazama, 2019). The author needs to address these facts to evaluate validity of these criteria at the very least, if he strongly wishes to use these criteria in his manuscript. Otherwise, the author must not use these "criteria".

*Reply: Thanks to the Reviewer #1 for the constructive comments to improve the quality of the manuscript. The method of Neuhäuser and Neuhäuser (2015) to classify the Aurora observation is removed from the manuscript and the table 2 is revised.*

**4. Solar Activity around 774/775**

While I appreciate scientific contribution by Mekhaldi et al. (2015) on the extreme solar storm in 774/775, Neuhäuser and Neuhäuser (2015) have claimed "they [their auroral records] cannot support a hypothetical solar super-flare" in page 236, for example. This is almost in an opposite spectrum against Mekhaldi et al. (2015). The author needs to clarify what he can say from Anatolian records for such scientific conflict.

*Reply: I would like to thank the Reviewer #1 for their thoughtful comments. The solar event of 774/775 by Mekhaldi et al. (2015) is removed from the manuscript.*

**5. Chronological Coverage**

Why "Any aurora observations could not be reached up to 1453"? That must be scientifically explained. Moreover, the title of "medieval Anatolian" should be revised to

"Byzantine" or "Byzantine Anatolian", given what the author surveyed.

*Reply: Because, 1453 is considered the end of the medieval period by historians. The title of the study is revised as "Historical Aurora Borealis Observations in medieval Anatolia (AD 1-1453): Implications for the past solar activity".*

**6. Definition of the Medieval Anatolia**

The author needs to see Figure 1 a little more carefully. Edessa (and probably Amida too) is/are situated outside of the Byzantine territory. More seriously, this figure explicitly shows that Constantinople is situated not in Asia Minor but in "Macedonia", while majority of the records in the author's catalog are derived from Constantinople. Therefore, Figure 1 shows that they are not in Anatolia either.

*Reply: The Figure 1 is revised according to the reviewer comments.*

**7. Relationship with Past Solar Activity**

While I commented on this aspect, the author just cited Willis and Stephenson (2001) without enough explanation. I cannot consider the author's addition as a "detailed explanation". Therefore, I have to repeat what I have written previously. The second conclusion in this manuscript states "In Anatolia and Middle East, there was a relatively high auroral activity during the years around 1100 is quite consistent with the naked-eye sunspot observations". However, the naked-eye sunspot observations are mentioned only briefly in in the context of Medieval Maximum (p.12) and periodicity between 1095 and 1204 is usual (Vaquero and Trigo, 2012). Therefore, the author should compare these auroral records with the naked-eye sunspot observations. Moreover, the cycle length during the Medieval Maximum is probably shorter (~9 years) on the basis of 14C data (Miyahara et al., 2008) and their cycle reconstructions are shown in Kataoka et al. (2017). Hence the existing statement for solar cycle length needs to be revised, citing Miyahara et al. (2008) and Kataoka et al. (2017). This enhanced solar activity is also better illustrated, citing the earliest datable sunspot drawing and relevant Korean auroral records in 1128 (Willis and Stephenson, 2001; Willis and Davis, 2014), and contrasted with the Oort Minimum (Usoskin et al., 2007, 2017; see also Inceoglu et al., 2015).

*Reply: I would like to thank the Reviewer #1 for their comments. The paragraph is revised according to the comments.*

**8. Relationship with Climatic Change**

As I commented previously, the logic was extremely difficult to follow and the revision of humidity with auroral record has been applied without scientific explanations. The author needs to seriously note that the relationship between solar activity and climatic change in historical time span is not very clear (Vaquero and Trigo, 2012; Lockwood et al., 2017). Lockwood et al. (2017) have especially clarified how misleading to explain the Little Ice Age with the Maunder Minimum. They have casted a caveat "The association of the solar Maunder minimum and the Little Ice Age is also not supported by proper inspection and ignores the role of other factors such as volcanoes" in page 2.23 for example. This made me strongly doubt the validity of the author's discussion for climatological impact. This manuscript cannot be published, unless otherwise the author removes their speculation about the climatic impact.

*Reply: I would like to thank the Reviewer #1 for the encouraging and constructive comments to improve the quality of the manuscript. The "Little Ice Age" is removed from the manuscript.*

**9. Conclusion**

Accordingly, the fifth and sixth conclusions must be removed, as well as the discussions on the climate change. In the same time, the author needs to clarify which made aurora visible in Anatolia so frequently in the Byzantine period: solar activity or intensity of dipole moment and position of geomagnetic pole.

*Reply: I would like to thank the Reviewer #1 for the constructive comments. The reason of the aurora in Anatolia so frequently is given in the "Results and Discussions" (second paragraph) and "Conclusions" (fourth conclusion) sections. On the other hand, the fifth and sixth conclusions are the important findings achieved from aurora observations besides historical-climatological data. Also, an additional conclusion is added.*

**_Response to comments from Anonymous Referee #2_:**

Title: "Historical Aurora Borealis Observations in Anatolia during medieval period (AD 1-1453): Implications for the past solar activity"
*Reply: I would like to thank the Reviewer #2 for the encouraging and constructive comments to improve the quality of the manuscript. The title of the manuscript is revised*

**General Comments**

In this paper, the author reports an overview of historical Aurora observations reports in Anatolia and Middle East regions in the medieval period based in historical texts, chronicles and aurora catalogs records. The paper presented a relationship between the auroral activity and the past solar activity, the past climatic changes, economy and society living in the remote time.
My view on the paper is that the discussions are interesting, and more discussions and clarifications were made in the revised version and the full paper desire publication in the ANGEO.
*Reply: I would like to thank the Reviewer #2 for the encouraging and constructive comments to improve the quality of the manuscript.*

**Major Comments:**

All the major comments in the previous version was answered in a proper way, and at this point I do not have any major recommendation.
*Reply: I would like to thank the Reviewer #2.*

**Minor issues:**

Line 1 (pg 1): In the title would not more correct to add the word "the" before "medieval period"? and also one space after "AD ..."?
*Reply: Revised*

Line 73 (pg 4): In the title of the section 2, I think that the word "Catalog" for the hAAc acronyms should be with "c" instead of "C".
*Reply: Revised*

Line 231 (pg 10): I suggest changing the sentence "23 different historical aurora records..." for "Twenty-three different aurora records...".
*Reply: Revised*

Line 357: In the Conclusions section, add the words "the" and "field" just after in the following sentence (check the correct language in this case): "... important information on variations in the geomagnetic field and auroral activity..."
*Reply: Revised*
The last suggestion is to enumerate the pages in the button right side or according to the journal standard.
*Reply: Revised*

We thank to you and the Reviewer #1 and Reviewer #2 for their constructive and helpful comments.

Sincerely,
Dr. Nafiz MADEN

[revised manuscript text omitted]

---

## Author Response (AR4)

Dear Dr. Igo Paulino,
Topical Editor
Annales Geophysicae (ANGEO)

**Ref**          **:** angeo-2019-97
**Title**        **:** Historical Aurora Borealis Observations in Anatolia during medieval
                   period: Implications for the past solar activity
**Journal**      **:** Annales Geophysicae (ANGEO)

Thank you for your constructive comments. I have just revised the manuscript in view of the reviewer comments as outlined in detail below and the paper is now ready to resubmit the journal of Annales Geophysicae (ANGEO) titled "Historical Aurora Borealis Observations in Anatolia during medieval period: Implications for the past solar activity". Please find our response to reviewer's comments step by step below.

I would like to thank the reviewers for their thoughtful comments. Responses to comments are presented in the following pages along with explanations.

Thanks again and looking forward to hearing from you soon.

Best regards,
**Dr. Nafiz MADEN**
Corresponding author

**Detailed Response to Reviewers**

**Response to comments from Anonymous Referee #1*:**

**General Comments:**

Apparently, the author has slightly improved this manuscript but the revision seems only stopgap and cannot resolve the problems at its roof. Therefore, I have addressed my concerns as positively as possible and tried to show how to revise this manuscript more explicitly. The novelty issue should be resolved pushing scientific implications for the medieval grand maximum and the extreme solar particle storm in 774.775, as the author does not have any novelty in the source records. The usage of "Anatolia" is highly problematic, as Constantinople, providing more than half of involved reports, is situated in the European side. The author needs to be more explicit about the obtained implication for the solar activity around the extreme solar particle storm in 774/775. The climatological discussions must be removed as the current evidence does not satisfy scientific threshold and will reduce the value of this manuscript. Overall, these comments are minimal requirements for publication in the Annales Geophysicae, which "publishes original articles and short communications (letters) on research of the Sun– Earth system...". The author is dully requested to address these comments appropriately and improve his scientific discussions and English grammar.

*Reply: I would like to thank the Reviewer #1 for encouraging comments to improve the quality of the manuscript. The manuscript is revised according to Reviewers comments. The title of the manuscript is changed as "Historical Aurora Borealis catalog for Anatolia and Constantinople (hAcAC) in the medieval period (AD 1-1453): Implications for the past solar activity" including observations recorded in the constantinople. Also, the sentence of "High auroral activity around the extreme solar particle storm in 774/775 and the medieval grand maximum in 1100s in Anatolia and Middle East is quite consistent with the past solar variability reported in other scientific literature" is added to the Abstract section.*

**Specific Comments**
**1. Novelty of the Records**
Unfortunately, compiling local auroral reports from existing catalogs does not guarantee novelty. This is especially the case, as the author explicitly admitted that he has not consulted the original historical documents and declined to provide example images of the original historical documents. The readers would have found its novelty, if the author extracted auroral records not from existing catalogs but from original historical documents.

Even more seriously, more than half of the auroral reports in this catalog (9 out of 14) are derived not from Anatolia but from Constantinople. As Constantinople is situated in the European side ("Macedonia" in the Byzantine Epoch or "Rumelia" in the Ottoman

Epoch), they are not classified as "historical Anatolian Aurora". This is explicitly shown in Figure 1. Therefore, this catalog must drop these 9 records, in order to let this manuscript be a "historical Anatolian Aurora catalog".

Therefore, the author needs to show its novelty in his scientific discussions, as these data are not new and more than half of them are not from Anatolia.

*Reply: Thank you for your encouraging comments to improve the quality of the manuscript. The novelty of this manuscript is given below:*
*The aim of this research is to establish a relationship between historical Aurora observations recorded in Anatolia and Constantinople during the medieval period and the past solar activity of interrelated social and economic climate change impacts. This research may also contribute to the understanding of public perception of the historical auroras. Anatolia and Constantinople have not been studied until now with respect to historical-climatological data and aurora observations. The available catalogs described above present a number of records covering Europe, Japan, China, Russia and Middle East regions.*
*The title of the manuscript is modified as "Historical Aurora Borealis catalog for Anatolia and Constantinople (hAcAC) in the medieval period (AD 1-1453): Implications for the past solar activity" covering aurora observations recorded in the constantinople.*
*High auroral activity around the extreme solar particle storm in 774/775 and the medieval grand maximum in 1100s in Constantinople, Anatolia and Middle East is quite consistent with the past solar variability reported in other scientific literature.*

**2. "Strength of the Aurora" and their Validity**

This has been much improved, removing misleading usage of likeliness evaluations. However, the narrative in L61-70 is then too long. Given the author's existing discussions, this lengthy phrase ("Neuhäuser and Neuhäuser (2015) are implemented ... cannot be classified as extreme events associated with extreme magnetic storms.") should be revised and connected more to their scientific discussions as follows:

"Recently, such candidate records of mid-latitude aurorae have been intensively investigated (e.g., Usoskin et al., 2013; Stephenson, 2015), due to the discovery of footprints of an extreme solar particle storm in the cosmogenic isotopes around 774/775 (Miyake et al., 2012; Usoskin et al., 2013; Mekhaldi et al., 2015). While Neuhäuser and Neuhäuser (2015) suggested five likeliness "criteria" and rejected most of the candidate aurorae around this event. However, these criteria actually contradicted auroral behaviour during the extreme space weather events (Kimball, 1960; Kataoka and Iwahashi, 2017; Kataoka et al., 2019; Kataoka and Kazama, 2019). Indeed, Stephenson et al. (2019) rejected these criteria and their analyses on the basis of multiple counter-examples during the extreme space weather events and confirmed an enhanced solar activity around this epoch. Their conclusion is consistent with the isotope evidence for the extreme solar particle storm such as the detected ratio of Be10 and Cl36 (Mekhaldi et al., 2015),

latitudinal concentration of C14 concentration (Uusitalo et al., 2018), and coincidental spikes of the multiple cosmogenic isotopes in both hemispheres (Büngten et al., 2018)."

*Reply: I would like to thank the Reviewer #1 for encouraging comments to improve this study. The paragraph is revised according to the Reviewer #1 suggestion.*

**4. Solar Activity around 774/775**

In order to push their scientific novelty, the author needs to expand this section, rather than dropping it. Extending what the author has written, I would suggest writing as follows, on the basis of what the author has claimed.

"The low-latitude aurorae of 772-773 are interesting, as being very close to the extreme solar event of 774/775 (Miyake et al., 2012; Usoskin et al., 2013; Mekhldi et al., 2015). These low-latitude aurorae are quite close from the extreme solar particle storm in 774/775 and support not the solar minimum (Neuhäuser and Neuhäuser, 2015) but high solar activity back then (Usoskin et al., 2013; Mekhaldi et al., 2015; Stephenson et al., 2019)."

*Reply: I would like to thank the Reviewer #1 for their thoughtful comments. The suggested paragraph is added to the manuscript.*

**5. Chronological Coverage**

To say "1453 is considered the end of the medieval period by historians", the authors must provide evidence. This is the end of Byzantine Empire, not the medieval epoch. I do not think the Ottoman conquest of the Constantinople is a benchmark of the medieval epoch. After all, it is not "medieval Anatolia" but "Byzantine Anatolia" that the author surveyed.

*Reply: The title of the manuscript is modified as "Historical Aurora Borealis catalog for Anatolia and Constantinople (hAcAC) in the medieval period (AD 1-1453): Implications for the past solar activity" covering aurora observations recorded in the constantinople.*

**6. Definition of the Medieval Anatolia**

The revision of Figure 1 can be mistaken as concealment, as Constantinople is anyways situated not in the Anatolian side but in the European side. The existing title with "Anatolia" is anyways highly misleading. If the author wishes to keep this title, the author must drop 9 records from Constantinople. The territory of Anatolia and modern Republic of Turkey is not the same.

*Reply: I would like to thank the Reviewer #1 for the encouraging and constructive comments to improve the quality of the manuscript. The title of the manuscript is modified as "Historical Aurora Borealis catalog for Anatolia and Constantinople (hAcAC) in the*

*medieval period (AD 1-1453): Implications for the past solar activity" covering aurora observations recorded in the constantinople.*

**7.* Relationship with Past Solar Activity and Climate Change**

As I commented before, the author must not mix up the solar activity and the terrestrial climate changes (see Vaquero and Trigo, 2012; Lockwood et al., 2017). As the author does not have a clear tie between the medieval solar maximum and medieval warm period, the author needs to discard almost everything between P12L269 and P14L309: "This study could also be significant constraints for exploration of solar activity on Earth's atmosphere and climate during the historical periods previously proved by Bard and Frank (2006). ... An important increase in agricultural production and population seems to have occurred in Anatolia after the year of 1100". If the author wishes to claim this relationship, he needs more supporting evidence and write another article.

Then, the author needs to rewrite his discussion on the medieval grand maximum focusing not on the periodicity but on the amplitude of solar cycles. I would suggest writing as follows.

"Vaquero and Trigo (2012) stated the period from 1095 to 1204 as an average solar cycle length, whereas this needs to be carefully compared with the reconstructed solar cycles on the basis of cosmogenic isotopes (Miyahara et al., 2008; Kataoka et al., 2017). Nevertheless, this period is characterised with numerous records of sunspots and aurorae shown in Vaquero and Vazquez (2009) and supported by Anatolian reports compiled in this article. This is highly consistent with an appearance of a gigantic sunspot in 1128 that caused a serious geomagnetic storm (Willis and Stephenson, 2001) and contrasts well with the Oort Minimum (Usoskin et al., 2007, 2017; see also Inceoglu et al., 2015). Indeed, Bekli et al. (2017) demonstrated that the naked-eye sun spot observations from 974 to 1278 and aurora records from 965 to 1273 show multiple unusual peaks related to the high solar activitiy at latitudes below 45° by using Chinese and Korean historical sources."

*Reply: I would like to thank the Reviewer #1 for their comments. The paragraph is revised according to the Reviewer comments.*

**9. Conclusion**

I appreciate that the author compiled auroral reports in Anatolia and Balkan Peninsula (Constantinople) from existing catalogs and compared them with other scientific results. However, unfortunately, what the author did in the climatological context does not satify scientific threshold and needs much more scientific supports. This can be done only writing another article for that issue. Therefore, the fifth and sixth conclusions must be removed, as well as the discussions on the climate change. Instead, the author should add their finding on the high solar activity around the extreme solar particle storm in 774/775 in the conclusion.

*Reply: I would like to thank the Reviewer #1 for the constructive comments. As I stated before the fifth and sixth conclusions are prominent findings achieved from aurora catalogs for Constantinople, Anatolia and Middle East regions.*

**Minor Comments**

P1L14-17: "High Aurora activity during the years around 1100 in Anatolia and Middle East is quite consistent with the past solar variability and planetary climatic changes drastically impacting on the economy and human events." => "High auroral activity around the extreme solar particle storm in 774/775 and the medieval grand maximum in 1100s in Anatolia and Middle East is quite consistent with the past solar variability reported in other scientific literature"

*Reply: Revised*

P7L145: Neuhäuser and Neuhäuser (2015) did not do anything more than Harrak (1999) for Zuqnin Chronicle. Just cite Harrak (1999). These Zuqnin records have been intensively analysed in Hayakawa et al. (2017). Cite it here and Table 2 #11.

*Reply: Revised*

P8L187: Cite references for definition of the Armenian years.

*Reply: Revised*

P14L313: "Medieval grand maximum" should not be mixed up with "Medieval Climate Anomaly".

*Reply: Revised*

We thank to you and the Reviewer #1 and Reviewer #2 for their constructive and helpful comments.

Sincerely,
Dr. Nafiz MADEN

---

## Author Response (AR5)

Dear Dr. Igo Paulino,
Topical Editor
Annales Geophysicae (ANGEO)

**Ref**         **:** angeo-2019-97
**Title**      **:** Historical Aurora Borealis Observations in Anatolia during medieval
                period: Implications for the past solar activity
**Journal**   **:** Annales Geophysicae (ANGEO)

       Thank you for your constructive comments. I have just revised the manuscript in view of the your comments as outlined in detail below and the paper is now ready to resubmit the journal of Annales Geophysicae (ANGEO) titled "Historical Aurora Borealis catalog for Anatolia and Constantinople (hABcAC) in the medieval period: Implications for the past solar activity". Please find our response to reviewer's comments step by step below.

       I would like to thank the reviewers for their thoughtful comments. Responses to comments are presented in the following pages along with explanations.

       Thanks again and looking forward to hearing from you soon.

       Best regards,
       **Dr. Nafiz MADEN**
       Corresponding author

**Detailed Response to Editor**

1. **Title:** Please see again the comment #5 from the Reviewer #2. Please revise Line 282 accordingly.
   *Reply: The title of the manuscript is revised as "Historical Aurora Borealis catalog for Anatolia and **Constantinople** (hABcAC) in the medieval period: Implications for the past solar activity".*

2. **Abstract:** Although Constantinople is close to Anatolia, it is not correct to use data from Constantinople to investigate changes in Anatolia, without mention that the data come from Europe. If you think that the European data are important to the conclusions, revise throughout the manuscripts and include Europe/Anatolia elsewhere you mention the data, including Tables and statements. I have read the mistake, at least in Lines 89--102 and Line 103. Tables including their caption must be revised as well.
   *Reply: The paragraph is revised*

3. **Introduction:** Please, revise the aim of this work and make sure that the conclusions form your work match with the goals. It is very important to be clear about the novelty of the present work during the Introduction section as well. Please, see again the comment #1 from the Reviewer #2.
   *Reply:The aim of this work is revised.*

4. **Figure 1:** See again the comment #4 from the Reviewer #2. I would recommend including in the Caption of Figure 1 that Constantinople was always on the European side.
   *Reply:  Figure and its caption are revised .*

5. **Conclusion:** Again, I agree with the Reviewer #2 and the conclusions must be fully revised. Please, keep only the findings from your work, considering your discussion and that you are sure that your work has figured out.
   *Reply: The conclusion section is revised.*

6. **Acknowledgements:** I would recommend to you keeping only the first statement. So, the Copernicus production will include statements about the reviewers and Editor.
   *Reply: The second sentence is deleted in the Acknowledgements section.*

We thank to you and reviewers for constructive and helpful comments.

Sincerely,
Dr. Nafiz MADEN

---

## Author Response (AR6)

Dear Dr. Igo Paulino,
Topical Editor
Annales Geophysicae (ANGEO)

**Ref**          **:** angeo-2019-97
**Title**        **:** Historical Aurora Borealis Observations in Anatolia during medieval
                 period: Implications for the past solar activity
**Journal**    **:** Annales Geophysicae (ANGEO)

Thank you for your constructive comments. I have just revised the manuscript in view of the your comments as outlined in detail below and the paper is now ready to resubmit the journal of Annales Geophysicae (ANGEO) titled "Historical Aurora Borealis catalog for Anatolia and Constantinople (hABcAC) in the medieval period: Implications for the past solar activity". Please find our response to reviewer's comments step by step below.

I would like to thank the reviewers for their thoughtful comments. Responses to comments are presented in the following pages along with explanations.

Thanks again and looking forward to hearing from you soon.

Best regards,
**Dr. Nafiz MADEN**
Corresponding author

**Detailed Response to Editor**

Comments to the Author:

Dear Dr. Nafiz Maden!

I sincerely ask apologies for the delay in accepting your manuscript. But, in my option, there are several points that need to be addressed before sending the manuscript for the production. I have revised the reviewer comments/suggestion for the previous round of revision and I have also found out that your scientific argumentation is really poor and there is no evidence for some statements that you have written. I would like you to do a final careful revision of the manuscript and attempt to address all the concerns listed below. Please, if you need extra time, do not hesitate to contact me, but do not submit the revision before be sure that you have solved all points.

*Reply:* Thank you for your constructive comments. I have just revised the manuscript in view of the your comments as outlined in detail below and the paper is now ready to resubmit the journal of Annales Geophysicae (ANGEO) titled "Historical Aurora Borealis catalog for Anatolia and Constantinople (hABcAC) in the medieval period: Implications for the past solar activity". Please find our response to comments step by step below.

**Major points:**

1. Figure 1: Remove the modern border. It is at best misleading and contradicts what the author claimed in the previous rebuttal letters.

    *Reply:The* modern *border in the Figure 1 is removed.*

2. Figure 2: Remove the first panel for humidity and third panel for agricultural development. They are not scientifically supported in this article and not relevant with their scientific results.

    *Reply: Figure 2 is revised.*

3. If the author still wishes to claim his result as a constraint for the medieval climatology, the author must show plot and visualize correlation of the local magnetic disturbance (with Turkish magnetograms) and local humidity or precipitation. Without such a figure based on the modern scientific data, the author's claims on climatology (L276-324) must be removed. Science must be developed based on scientific data and scientific evidence.

    *Reply:The climate change interpretations are removed (L276-324).*

4. The author must clarify that they have not consulted the original historical documents but only compiled the existing catalogs. There are no originalities for these records, as the author has not conducted an original investigation. This must be clarified.

*Reply: The aim of this research is to establish a relationship between historical Aurora observations recorded in Anatolia and Constantinople during the medieval period and the past solar activity. Anatolia and Constantinople have not been studied until now with respect to historical-climatological data and aurora observations. The available catalogs present a number of records covering Europe, Japan, China, Russia and Middle East regions. This research may also contribute to the understanding of public perception of the historical auroras.*

**Minor points:**

1. The title should be revised as 'medieval' => 'Byzantine'. There are no Ottoman records. The author has only investigated Byzantine reports, not medieval Anatolia and Constantinople.

   *Reply: The title of the manuscript is revised as "Historical Aurora Borealis catalog for Anatolia and Constantinople (hABcAC) during the Byzantine period: Implications for the past solar activity."*

2. P1L7-8: In this paper, it is reviewed the relationships between the aurora observations, past solar activity and climatic change in Anatolia during the medieval period. => In this paper, Anatolian aurora has been reviewed based on the existing catalogs. [NB the author has not conducted an original survey but compiled existing published catalogs.]

   *Reply: The sentence is revised as "Herein, Anatolian aurora has been reviewed based on the existing catalogs to establish a relationship between the aurora observations and past solar activity during medieval period."*

3. For this purpose, it is presented two historical aurora catalogs for Constantinople, Anatolia 9 and Middle East regions at various dates by using historical texts, chronicles and other 10 auroral records. => For this purpose, historical aurora catalogs for Constantinople and Anatolia are compiled based on the existing catalogs and compared with those in Middle East regions. [NB, Again, the authors have not consulted historical documents]

   *Reply:Revised*

4. P3L64-65: However, these criteria actually contradicted auroral behaviour during the extreme space weather events (Kimball, 1960; Kataoka and Iwahashi, 2017; Kataoka et al., 2019; Kataoka and Kazama, 2019). => However, these criteria directly contradicted auroral behaviour during the extreme space weather events, as overhead aurora can extend down to ~25° in magnetic latitude (vs 40 – 50° in Anatolia) and the whitish aurora

appears more equatorial side (Kimball, 1960; Kataoka and Iwahashi, 2017; Kataoka et al., 2019; Kataoka and Kazama, 2019).

*Reply:Revised*

5. P4L77-79: The goal of this study is to compile a historical aurora catalog to analyse the past solar activity of interrelated social, economic and climate change impacts during the medieval period. => The goal of this study is to compile a historical aurora catalog based on the existing catalogs, in order to analyse the past solar activity during the medieval period.

*Reply:Revised*

6. P4L80-81: Constantinople and Anatolia have not been studied up to now with regard to historical-climatological data and aurora observations. => Constantinople and Anatolia have only been peripherally discussed up to now with regard to auroral observations. [NB: the author has compiled this catalog based on the existing catalogs. So previous scholars have already known these aurorae.]

*Reply:Revised*

7. P6L133: Harrak (1999) listed two aurorae records => Harrak (1999) and Hayakawa et al. (2017) listed two aurorae records.

*Reply:Revised*

8. P6L135, P7L141, P12L269, and P15L348: Amida (Turkey) => Amida

*Reply:Revised*

9. P10L225: Constantinople and Anatolia during the medieval period. => Constantinople and Anatolia during the medieval period based on the existing catalogs.

*Reply:Revised*

10. P11L235-236: They are the longest direct observational records available for studying solar and space weather dynamics. => That's not true. What about the cosmogenic isotopes? Just remove it.

*Reply:Removed.*

11. P12L276-P14L311: Discussions on climatology => Remove it. Not supported by scientific evidence.

*Reply:Removed.*

12. P14L316-324: Remove it. Not supported by scientific evidence.

*Reply:Removed.*

13. Table 1: Sources => Existing catalogs

*Reply:Revised.*

**14.** Table 3: Giving a spot coordinate for Asia Minor is highly misleading. Remove it.

*Reply:Revised.*

**15.** Table 4: Reports 3 – 6 are not from the Middle East but from Byzantium. Remove them.

*Reply:Removed.*

**16.** Table 5: This is irrelevant to what the author has scientifically shown. Remove it.

*Reply:Removed.*

We thank to you and reviewers for constructive and helpful comments.

Sincerely,
Dr. Nafiz MADEN